# IPod: Inverse-Problem-Driven Meta-Learning for Fast Generalizable Neural Representations in MRI Reconstruction

## Abstract

Implicit neural representation (INR) demonstrates strong performance in magnetic resonance imaging (MRI) reconstructions by learning continuous mappings from spatial coordinates to signal intensities. However, existing unsupervised INR approaches require training from scratch for each observation, which is time-consuming and limits practical deployment. In this work, we propose an inverse-problem-driven meta-learning framework (IPOD) that learns generalizable parameter initializations for INR directly from various undersampled reconstruction tasks without requiring fully sampled references. Technically, the meta-update is adaptively modulated by the hyperparameter performance of each inverse problem, ensuring optimal parameter distributions for robust and efficient initialization. Our approach leverages diverse reconstruction tasks with varying sampling patterns and anatomical structures to acquire a powerful and robust prior. Experimental validations demonstrate that the proposed framework provides powerful initialization that achieves fast convergence and superior reconstruction quality across different imaging protocols, outperforming baseline INR methods. Furthermore, this framework eliminates the dependence on reference images in conventional meta-learning procedures and has the potential to be extended to INR-based solutions for a wide range of imaging inverse problems. The code and data will be available at: https://anonymous.4open.science/r/iPod-2C60

## 1 Introduction

Magnetic resonance imaging (MRI) offers superior soft tissue contrast, making it essential for clinical diagnosis and neuroscience research (Feng et al., 2023). However, its long acquisition times lead to higher healthcare costs and patient discomfort. Accelerating MRI acquisition is commonly achieved by collecting undersampled $k$-space data, but reconstructing high-quality images under such conditions is challenging due to violations of the Nyquist sampling theorem. With the advancement of deep learning (DL), supervised reconstruction methods have shown remarkable improvements by learning mappings from corrupted to artifact-free MR images (Sun et al., 2020; Han et al., 2018; Aggarwal et al., 2019; Qin et al., 2019; Wang et al., 2022; 2023b;a). Nevertheless, these approaches require large-scale fully sampled data, which is impractical, and often suffer from poor generalization under domain shifts at inference.

Recently, implicit neural representations (INRs) have emerged as an unsupervised paradigm to solve inverse problems in medical imaging (Shen et al., 2022; Xu et al., 2023; Spieker et al., 2023). In accelerated MRI reconstruction, INR-based methods represent the desired MR image as a continuous function and integrate the physical forward model to achieve comparable performance with supervised methods using only undersampled $k$-space data. However, existing INR approaches face two key limitations: unstable performance across high acceleration factors, and slow reconstruction speeds—particularly in 3D reconstruction scenarios, where reconstruction may take several hours.

Previous works have demonstrated that efficient initialization enables faster INR convergence on unseen data, with meta-learning emerging as a fundamental framework for learning optimal initialization strategies across multiple tasks (Bauer et al., 2023; Dupont et al., 2022; Tancik et al., 2021). However, these meta-initialized INRs are predominantly applied to computer vision tasks such as

image regression and novel view synthesis. For medical image reconstruction inverse problems from undersampled raw measurments, the potential of meta-learning-based INR initialization remains unexplored. The unique challenges in medical imaging, including the scarcity of high-quality training data and the critical need for integration of physics-based forward models, have hindered the direct application of existing meta-learning approaches to this domain.

In this work, we propose **IPOD**, an inverse-problem-driven meta-learning framework that learns effective initialization of INR networks for MRI reconstruction. Our key innovation lies in constructing a novel meta-learning framework to efficiently meta-initialize INR networks without the dependency on high-quality fully-sampled data. Conceptually, leveraging the physical forward model, our approach transforms the inner loop optimization to focus on solving diverse reconstruction inverse problems rather than simple image regression tasks. This physics-informed meta-learning strategy enables the framework to capture the real-world reconstruction processes, leading to more informed and robust initialization parameters. Furthermore, we introduce an adaptive weighting mechanism that prevents the meta-learning process from being corrupted by suboptimal solution spaces. This mechanism dynamically ensures the stability and effectiveness of meta-initialized INR networks across diverse imaging scenarios and acceleration factors.

In the experiments, we adopt fastMRI (Knoll et al., 2020) dataset for IPOD meta-learning training phase, and comprehensively evaluate on the out-of-domain datasets, including public MoDL (Aggarwal et al., 2018) dataset and prospectively undersampled in-house data. The results consistently demonstrate rapid convergence across diverse out-of-domain scenarios, including different subjects, contrast mechanisms, sampling patterns, and forward models. Notably, IPOD is a unified meta-learning framework that provides effective initialization for any INR method. To validate its effectiveness, we evaluate it on three representative INR frameworks. The main contributions are summarized as follows:

- We propose a novel meta-learning paradigm that eliminates the dependency on fully-sampled reference images by learning effective INR initialization directly from corrupted datasets.

- We integrate physics-based forward models into the meta-learning process, transforming the inner loop optimization from simple image fitting to solving diverse reconstruction inverse problems.

- We conduct extensive experiments demonstrating that IPOD can offer powerful initialization for diverse INR models, enabling faster convergence and improved reconstruction quality across varying MRI reconstruction scenarios.

## 2 RELATED WORKS

**INR for MRI Reconstruction** Implicit Neural Representation (INR) has emerged as a powerful self-supervised deep learning framework for accelerated MRI reconstruction (Feng et al., 2023; Shen et al., 2022; Xu et al., 2023; Kunz et al., 2023; Feng et al., 2022; Spieker et al., 2023; Wu et al., 2021; Huang et al., 2023; Chen et al., 2023; Catalán et al., 2023), with approaches ranging from k-space signal prediction to direct image intensity reconstruction. It represents the desired MR image as a continuous function of spatial coordinates, approximated by a neural network. By integrating the MRI physical forward model and the network's learning bias (Rahaman et al., 2019; Xu et al., 2019), INR can reconstruct high-quality MR images from only undersampled $k$-space data. However, most INR approaches rely on scan-specific training, which fails to exploit valuable data-driven priors. This limitation is a primary reason for their unstable performance and relatively slow reconstruction speeds, as each new dataset requires training from scratch, and the learned representations are difficult to transfer even to similar data domains. In contrast, we propose a promising direction: integrating data-driven priors via meta-learning. This approach enables us to find a well-initialized INR, which significantly enhances both reconstruction quality and speed.

**Meta-Learning** Meta-learning addresses fast adaptation and generalization with few samples, where a meta-learner is trained to quickly adapt to new tasks (Sung et al., 2018; Vinyals et al., 2016; Ravi & Larochelle, 2017; Snell et al., 2017; Mishra et al., 2018). The popular branches are optimization-based meta-learning methods like MAML (Finn et al., 2017) and Reptile (Nichol et al.,

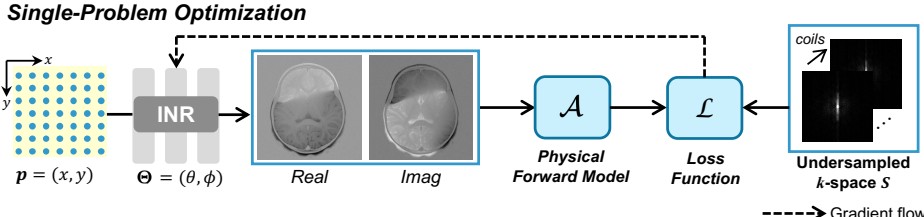

Figure 1: Given a single undersampled $k$-space data $S$, the INR networks, parametrized by $\Theta = (\theta, \phi)$, take discrete coordinates $x = (x, y)$ as input and produce the predicted real and imaginary values of the underlying MR image. Based on the forward physical model, the loss function $\mathcal{L}$ is minimized for optimization of network parameters $\Theta$.

2018), which aim to find strong weight initializations that allow efficient adaptation to unseen tasks within a few optimization steps. Recent advancements have connected meta-learning with INRs, extending its possibilities to learn functions that represent data (Sitzmann et al., 2020a; Tancik et al., 2021). Moreover, the concept of using meta-learning for INR initialization has been explored in various works (Lee et al., 2021; Chen & Wang, 2022; Vyas et al., 2024; Lee et al., 2023; Kim et al., 2023; Gu et al., 2023; Yang et al., 2025). However, these methods are difficult to directly apply to inverse problems in MRI reconstruction. First, most of them require large amounts of high-quality training data, which is challenging to collect in medical imaging. Second, they typically lack integration of the physical forward model, which is essential for simulating real-world data acquisition processes and ensuring high-fidelity MRI reconstruction.

## 3 PROPOSED METHOD

### 3.1 PRELIMINARIES

**MRI Inverse Problem**   Multi-coil MRI reconstruction aims to recover an unknown MR image from under-sampled $k$-space measurements, which is typically an ill-posed problem due to the violation of the Nyquist sampling theorem. The under-sampled $k$-space data $S_j$ can be written as:

$$S_j = \mathbf{M}\mathcal{F}\mathbf{C}_j I + e_j, \quad j \in \{1, 2, \ldots, N\}, \tag{1}$$

where $I$ denotes the desired MR image, $e_j$ is the measurement noise of the $j$-th coil, $N$ is the number of coils, $\mathbf{C}_j$ denotes a diagonal matrix representing the $j$-th coil sensitivity map, $\mathcal{F}$ is the Fourier transform matrix, and $\mathbf{M}$ is the sampling mask. For simplicity, we omit the coil subscript $j$ in the following derivations. Defining $\mathcal{A} = \mathbf{M}\mathcal{F}\mathbf{C}$ as the physical forward model, the inverse problem can be formulated as:

$$I^* = \arg\min_I \quad \frac{1}{2}\|S - \mathcal{A}I\|_2^2 + \lambda \cdot \|GI\|_1, \tag{2}$$

where $G$ denotes the gradient operator enforcing smoothness regularization, and $\lambda$ is a hyper-parameter controlling its contribution. The key to addressing this ill-posed problem is the design of a reliable prior that effectively constrains the solution space, thus enabling desired image reconstructions. Since the explicit total variational regularizer cannot fully capture the complex distribution of MR images, its performance remains limited.

**Conventional INR for MRI Reconstruction**   Implicit neural representation (INR) formulates the complex-valued MR image $I$ as a continuous function of spatial coordinates (Feng et al., 2022; 2023), which are parametrized by two separate MLP networks as follows:

$$I(p) = f_{\text{real}}(p) + i \cdot f_{\text{imag}}(p), \tag{3}$$

where $p = (x, y)$ denotes 2D spatial coordinates in the normalized imaging space $\Omega = [-1, 1] \times [-1, 1]$. The functions $f_{\text{real}} : \mathbb{R}^2 \to \mathbb{R}$ and $f_{\text{imag}} : \mathbb{R}^2 \to \mathbb{R}$ correspond to MLPs representing the real and imaginary components, respectively, with $\theta$ and $\phi$ as their learnable parameters.

As illustrated in Fig. 1, INR optimizes the two networks $f_{\text{real}}$ and $f_{\text{imag}}$ to recover high-quality MR images in an unsupervised manner. Specifically, the networks take all coordinates $p \in \Omega$ as input

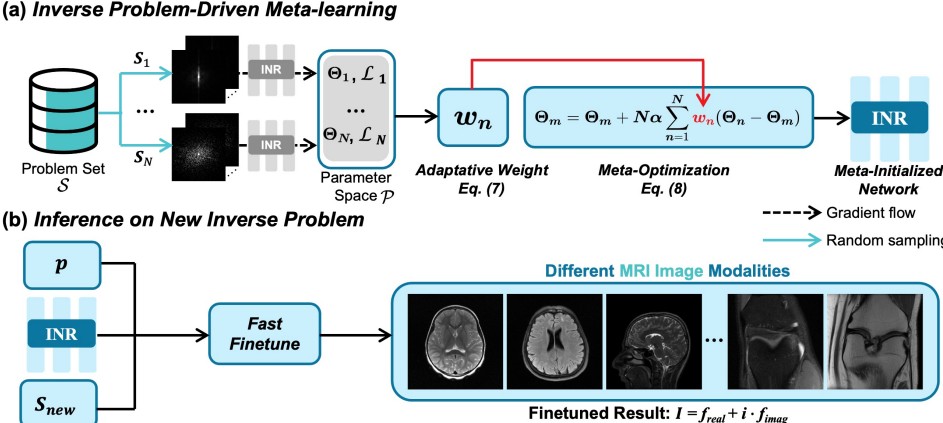

Figure 2: Overview of IPOD framework. In the inverse-problem-driven meta-learning procedure, a subset is first randomly sampled from the task set $\mathcal{S}_{\text{sub}} = \{\boldsymbol{S}_1, \boldsymbol{S}_2, \ldots, \boldsymbol{S}_N\} \subseteq \mathcal{S}$. After completing optimization, as shown in Fig. 1, for each single task $\boldsymbol{S}_n \in \mathcal{S}_{\text{sub}}$, adaptive weights are computed, which are used to generate efficient optimization directions in meta-optimization $\boldsymbol{\Theta}_m$. Ultimately, these meta-initialized INR networks are capable of performing fast and high-quality reconstruction across various MRI inverse problems.

and predict the corresponding real part $\boldsymbol{f}_{\text{real}}(\boldsymbol{p})$ and imaginary part $\boldsymbol{f}_{\text{imag}}(\boldsymbol{p})$. The predicted image $\boldsymbol{I}$ is then transformed into $k$-space estimations $\mathcal{A}\boldsymbol{I}$. Since the forward model $\mathcal{A}$ is differentiable, the networks can be optimized using gradient-based backpropagation. Consequently, $\boldsymbol{f}_{\text{real}}$ and $\boldsymbol{f}_{\text{imag}}$ are jointly trained by minimizing the loss function $\mathcal{L}$, which consists of a data consistency and a smoothness regularizer. Formally, INR solves the following optimization:

$$\boldsymbol{\Theta}^* = \arg\min_{\boldsymbol{I}} \quad \frac{1}{2}\|\boldsymbol{S} - \mathcal{A}\boldsymbol{I}\|_2^2 + \lambda \cdot \|\boldsymbol{G}\boldsymbol{I}\|_1,$$
$$\text{subject to} \quad \boldsymbol{I}(\boldsymbol{p}) = \boldsymbol{f}_{\text{real}}(\boldsymbol{p}) + i \cdot \boldsymbol{f}_{\text{imag}}(\boldsymbol{p}).$$
(4)

where $\boldsymbol{\Theta}^* = (\theta^*, \phi^*)$ denotes the optimal parameters. Benefiting from the inherent inductive bias of neural networks toward continuous image structures, i.e., spectral bias (Rahaman et al., 2019), the INR parameterization enables accurate approximation of the underlying continuous function. After optimization, feeding all coordinates $\boldsymbol{p}$ into the trained networks $\boldsymbol{f}_{\text{real}}^*$ and $\boldsymbol{f}_{\text{imag}}^*$ yields the final high-quality MR image reconstruction $\boldsymbol{I}^*(\boldsymbol{p}) = \boldsymbol{f}_{\text{real}}^*(\boldsymbol{p}) + i \cdot \boldsymbol{f}_{\text{imag}}^*(\boldsymbol{p})$.

**Our Motivation** Although existing INR-based methods for the MRI reconstruction task demonstrate great potential, their scan-specific optimization faces two key limitations: 1) they fail to exploit population-level data priors, limiting performance for diverse cases; 2) they start optimization from randomly initialized networks, reducing convergence stability and speed.

To this end, we aim to learn effective initializations for INR networks, thereby enabling rapid and robust adaptation to diverse MRI scenarios. Specifically, we propose an inverse-problem-driven framework, namely IPOD, which can learn generalized and robust INR initializations directly from population-level undersampled $k$-space data *without* the need for any high-quality MR images. Our key insight is that diverse MRI inverse problems share underlying similarities in their solution manifolds. By learning from a population of inverse problems during meta-training, IPOD captures common patterns and embeds them into initialization parameters. These population-level priors provide an informed starting point, enabling initialized INRs to generalize effectively across different imaging scenarios with varying anatomy, contrast mechanisms, and sampling patterns.

### 3.2 INVERSE PROBLEM-DRIVEN META-LEARNING

Fig. 2 a shows the pipeline of INR initializations by the proposed IPOD. Technically, IPOD adapts the Reptile (Nichol et al., 2018) algorithm to meta-initialize both $\boldsymbol{f}_{\text{real}}$ and $\boldsymbol{f}_{\text{imag}}$ across multiple inverse reconstruction problems, eliminating the need for fully-sampled $k$-space data while lever-

aging physics-based forward models. It consists of three main components, which we detail in the following sections.

**Inverse Problem Set Construction**    To enable effective meta-learning across diverse MRI reconstruction scenarios, we construct a comprehensive inverse problems set $\mathcal{S} = \{\boldsymbol{S}_1, \boldsymbol{S}_2, \ldots, \boldsymbol{S}_L\}$ comprising undersampled various $k$-space. To ensure balanced representation and mitigate bias, we systematically curate the task set to include different undersampling ratios, various $k$-space trajectories, and multiple image categories. This diversity enables our framework to learn generalizable initialization strategies that are robust across different MRI applications.

**Problem-Specific Inner Loop**    Within each inner loop iteration, an undersampled $k$-space data is first sampled from the task set, i.e., $\boldsymbol{S}_n \in \mathcal{S}$, and the INR-based reconstruction process is conducted to solve the corresponding inverse problem like the optimization procedure in Fig. 1. Specifically, given the sampled data $\boldsymbol{S}_n$, we use the corresponding INR network $\boldsymbol{f}_{\text{real}}^n$ and $\boldsymbol{f}_{\text{ima}}^n$ with learnable paramter $\boldsymbol{\Theta}_n$ to predict the underlying MR images $\boldsymbol{I}_n(\boldsymbol{p}) = \boldsymbol{f}_{\text{real}}(\boldsymbol{p}) + i \cdot \boldsymbol{f}_{\text{imag}}(\boldsymbol{p})$. Then, we use the physical forwar model $\mathcal{A}_n = \mathbf{M}_n \mathcal{F} \mathbf{C}_n$ specficed to the measuremnt $\boldsymbol{S}_n$ for generating the $k$-space estimate $\mathcal{A}_n \boldsymbol{I}_n$. Finally, we calculate the problem-specific loss $\mathcal{L}_n$, which is calculated as:

$$\mathcal{L}_n(\boldsymbol{\Theta}_n, \boldsymbol{S}_n) = \frac{1}{2}\|\boldsymbol{S}_n - \mathcal{A}_n \boldsymbol{I}_n\|_2^2 + \lambda \cdot \|\mathbf{G}\boldsymbol{I}_n\|_1. \tag{5}$$

These parameters $\boldsymbol{\Theta}_n$ are iteratively optimized via gradient descent algorithms to minimize the physics-informed loss function, with updates computed as:

$$\boldsymbol{\Theta}_n \leftarrow \boldsymbol{\Theta}_n - \gamma \nabla_{\boldsymbol{\Theta}_n} \mathcal{L}_n(\boldsymbol{\Theta}_n, \boldsymbol{S}_n), \tag{6}$$

where $\gamma$ denotes the inner loop learning rate. This optimization process adapts the INR parameters to the specific characteristics of each inverse problem, and the resulting problem-adapted parameters $\boldsymbol{\Theta}_n$ are subsequently utilized in the meta-update phase.

**Adaptive Weighted Meta-optimization**    In each meta-learning epoch, we first randomly sample a subset from the task set $\mathcal{S}_{\text{sub}} = \{\boldsymbol{S}_1, \boldsymbol{S}_2, \ldots, \boldsymbol{S}_N\} \subseteq \mathcal{S}$. For each task $\boldsymbol{S}_n \in \mathcal{S}_{\text{sub}}$, the problem-adapted parameters $\boldsymbol{\Theta}_n$ are updated through inner-loop optimization (Eq. 6). After completing all inner loops, we obtain a parameter space $\mathcal{P} = \{(\boldsymbol{\Theta}_1, \mathcal{L}_1), (\boldsymbol{\Theta}_2, \mathcal{L}_2), \ldots, (\boldsymbol{\Theta}_N, \mathcal{L}_N)\}$, where each pair represents problem-specific parameters and the corresponding reconstruction loss. However, severely ill-posed inverse problems may lead to suboptimal parameter learning due to insufficient constraints and ambiguous solution spaces. Specifically, when confronted with challenging reconstructions, the optimization may converge to poor local minima or exhibit unstable convergence behavior, resulting in network parameters that inadequately represent the underlying image structure. Directly incorporating such parameters into meta-updates can degrade the overall initialization.

To alleviate such potential negative effect, we introduce an adaptive weighting strategy that selectively emphasizes inverse problems with better solution sapce. Specifically, each task is assigned a weight based on the inverse of its loss:

$$\boldsymbol{w_n} = \frac{1/(\mathcal{L}_n + \epsilon)}{\sum_{n=1}^N 1/(\mathcal{L}_n + \epsilon)}, \tag{7}$$

where $\epsilon = 1 \times 10^{-8}$ prevents division by zero. The meta-parameters $\boldsymbol{\Theta}_m$ is then updated using a weighted variant of Reptile (Nichol et al., 2018):

$$\boldsymbol{\Theta}_m \leftarrow \boldsymbol{\Theta}_m + N\alpha \sum_{n=1}^N \boldsymbol{w_n}(\boldsymbol{\Theta}_n - \boldsymbol{\Theta}_m). \tag{8}$$

This mechanism reduces the impact of poorly-performing tasks and yields robust initializations that generalize across diverse sampling patterns and anatomical structures. After all updates, we obtain the meta-initialized INR networks, which serve as efficient, robust, and broadly generalizable initializations, facilitating reliable adaptation to diverse undersampled MRI reconstruction scenarios.

Table 1: Quantitative results (Mean ± STD in PSNR) comparing different INR baselines with and without proposed **IPOD** initialization on different datasets, anatomies, sampling patterns and physical forward models. The best performance for each method type is highlighted in **bold**.

| Dataset | Sampling | AF | DINER | | SIREN | | HASH | |
|---|---|---|---|---|---|---|---|---|
| | | | w/o IPOD | w/ IPOD | w/o IPOD | w/ IPOD | w/o IPOD | w/ IPOD |
| Brain (fastMRI) | Cartesian | 3 | 58.58±2.92 | **58.98±3.14** | 44.50±3.97 | **55.24±5.34** | **60.68±4.91** | 59.88±4.41 |
| | | 4 | 48.61±5.12 | **48.89±4.87** | 37.12±3.62 | **41.61±5.16** | 47.32±3.37 | **47.78±3.25** |
| | Radial | 10 | 36.73±3.79 | **39.69±1.66** | 32.73±3.00 | **37.23±3.03** | 38.95±2.44 | **39.26±2.12** |
| | | 14 | 33.41±4.08 | **36.87±1.78** | 31.08±2.42 | **34.94±1.65** | 35.99±2.04 | **36.21±2.42** |
| Knee (fastMRI) | Cartesian | 3 | 51.20±5.11 | **54.24±4.01** | 36.06±2.07 | **43.45±3.95** | 54.21±5.87 | **56.63±2.16** |
| | | 4 | 39.25±3.95 | **40.33±2.32** | 35.96±1.71 | **38.86±1.81** | 39.31±2.30 | **40.50±1.89** |
| | Radial | 10 | 33.76±2.66 | **35.14±1.60** | 30.13±2.30 | **33.30±1.50** | 34.94±1.62 | **34.96±1.61** |
| | | 14 | 31.25±3.21 | **33.46±1.48** | 29.21±1.79 | **31.26±1.50** | 33.03±1.71 | **33.17±1.61** |
| Brain (MoDL) | Cartesian | 3 | 49.80±4.18 | **53.93±4.51** | 52.24±1.39 | **59.18±1.89** | 57.60±3.09 | **59.36±2.65** |
| | | 4 | 40.08±3.55 | **52.13±4.62** | 41.77±1.11 | **53.69±1.55** | 56.71±3.21 | **58.07±2.40** |
| | Radial | 10 | 32.15±2.04 | **37.76±2.68** | 30.31±0.52 | **34.46±3.60** | 37.54±1.51 | **37.72±0.78** |
| | | 14 | 29.46±2.13 | **35.75±0.49** | 29.88±0.61 | **32.56±0.59** | 34.69±0.80 | **35.00±1.18** |

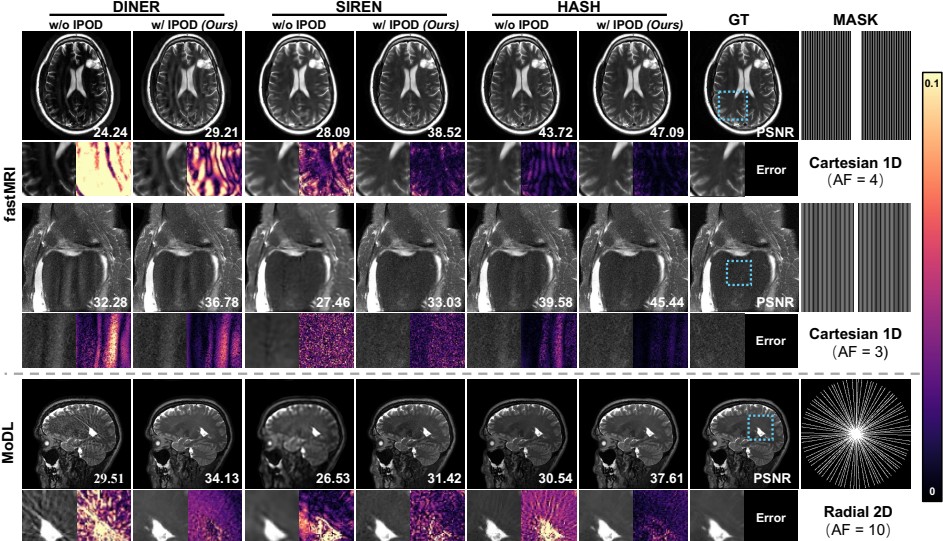

Figure 3: Qualitative and quantitative comparison of baselines with and without IPOD initialization under a small number of parameter updates (150 iterations) on fastMRI and MoDL datasets.

### 3.3 INFERENCE ON UNSEEN UNDERSAMPLED $k$-SPACE DATA

Once meta-training is completed, the meta-initialized parameters $\Theta_m$ are employed as the starting point to finetune INR networks on unseen undersampled $k$-space data. As shown in Fig. 2 b, the finetuning procedure follows the same adaptation scheme as the inner-loop updates during meta-training, where $\Theta_m$ is refined through several optimization iterations. Importantly, the meta-initialized INRs are not restricted to the inverse problem categories seen during meta-training, but can effectively generalize to undersampled data drawn from broader and unseen data distributions, enabling robust reconstructions across diverse MRI scenarios.

## 4 EXPERIMENTS

### 4.1 EXPERIMENTAL SETTINGS

**Datasets** The retrospective experiments are conducted on fastMRI (Knoll et al., 2020) and MoDL datasets (Aggarwal et al., 2018). For meta-training, we extract 3,600 undersampled $k$-space mea-

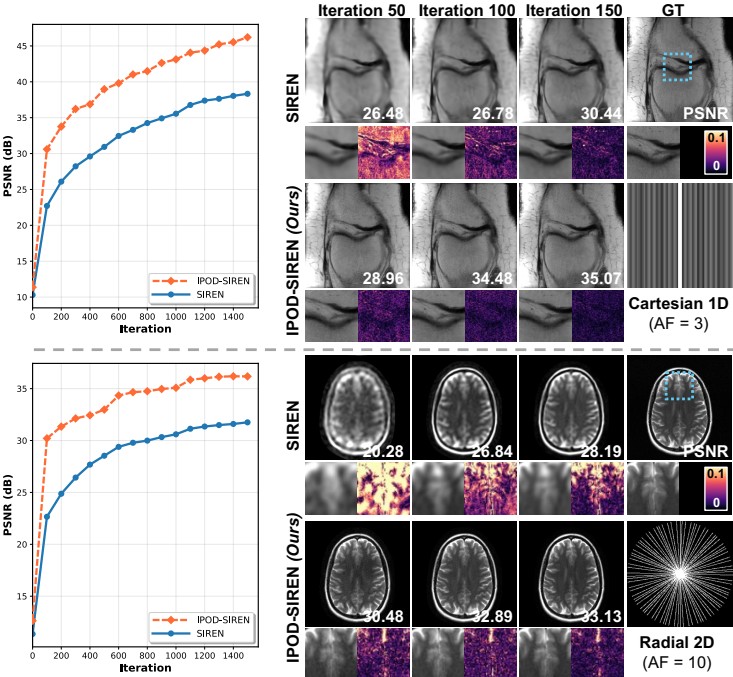

Figure 4: Comparison of performance curves and qualitative/quantitative results of SIREN with and without IPOD initialization under varying parameter update iterations on different datasets.

surements from fastMRI, encompassing different contrasts (T2w, FLAIR), anatomical structures (brain, knee) with $256 \times 256$ resolution. For evaluation, we test on unseen scenarios including different subjects and contrasts. Specifically, we use 50 brain slices (T1w, T2w, FLAIR) and 50 knee slices from the fastMRI dataset, plus 50 T2w brain slices with $256 \times 256$ from 20 subjects in the MoDL dataset. Additionally, we acquire one T1w brain 3D scan from a 3.0 T United Imaging Healthcare (UIH) uMR 890 scanner for prospective real-world data validation.

**Pre-processing** For meta-learning training data, we employ two distinct sampling patterns: 1D Cartesian undersampling and 2D random undersampling. The 1D Cartesian sampling encompasses acceleration factors (AFs) of 2, 4, and 6, while the 2D random sampling includes AFs of 5, 10, and 15. To validate the generalization of initialization parameters, we additionally implement three unseen sampling patterns: 2D Poisson sampling, 2D Gaussian sampling, and 2D radial sampling with golden-angle acquisition scheme. *More details about pre-processing can be found in the Appendix.*

**Baselines and Metrics** We select three representative INR models as baselines: DINER (Xie et al., 2023), SIREN (Sitzmann et al., 2020b), and HASH (Müller et al., 2022), covering methods with nonlinear activation functions and MLP structures with coordinate encoders. To evaluate IPOD's effectiveness, we apply our meta-learning framework to each baseline, creating IPOD-DINER, IPOD-SIREN, and IPOD-HASH variants. We compare these meta-initialized models against their random initialization strategies to demonstrate IPOD's universal applicability and performance improvements. For the reconstructed MR images, we use peak signal-to-noise ratio (PSNR) and structural similarity index (SSIM) as quantitative evaluation metrics. *More implementation details can be found in the Appendix.*

**Implementation Details** All methods are performed on PyTorch. For the IPOD meta-learning phase, we construct the set $\mathcal{S}$ of inverse problems with 72 different types, where each type contains 50 samples from different subjects, resulting in $L = 3,600$ total undersampled $k$-space data. We set the batch size $N$ in each meta-learning epoch to 15, and the number of meta-learning training epochs to 2,500 with a meta-learning rate $\alpha = 5 \times 10^{-4}$. For inner loop optimization, we set different learning rates for each baseline: $\gamma = 2 \times 10^{-2}$ for DINER and HASH, $\gamma = 2 \times 10^{-4}$

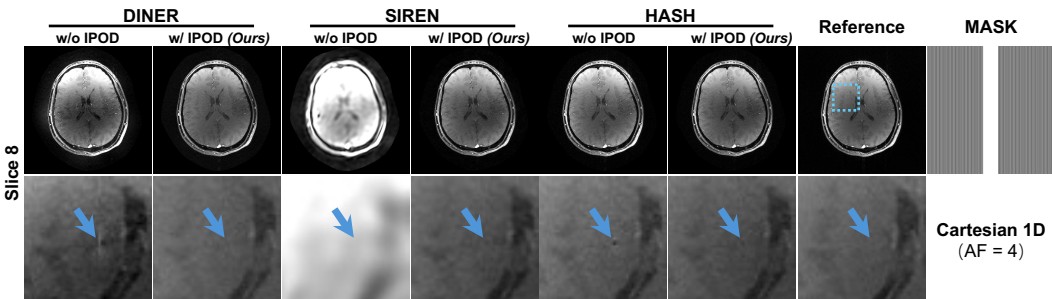

Figure 5: Qualitative comparison of baselines with and without IPOD initialization under a small number of parameter updates (75 iterations) on prospectively undersampled real-world data.

for SIREN during meta-learning. The total variation regularization weight is fixed to $\lambda = 2$ across all datasets. We use the Adam algorithm with default hyper-parameters (Kingma & Ba, 2014) to optimize all the models. Notably, two types of physical forward models are employed: the fast Fourier transform (FFT) and Non-uniform Fast Fourier Transform (NuIFFT) (Fessler, 2007). All experiments were performed on a single NVIDIA RTX A100 GPU. *Due to the space constraint, we introduce the other implementation details of IPOD and baselines in the Appendix.*

## 4.2 MAIN RESULTS

**Performance-Oriented Comparison with Baselines**   Table 1 presents quantitative PSNR results for different INR architectures with and without IPOD initialization. IPOD consistently improves reconstruction quality across all baselines, datasets, and sampling patterns, with particularly pronounced gains under challenging conditions such as higher acceleration factors and radial sampling. Among all architectures, SIREN benefits most from IPOD initialization.

**Efficiency-Oriented Comparison with Baselines**   To demonstrate the fast generalization ability of IPOD-initialized networks on unseen data, we conducted an efficiency comparison with baselines using only 150 parameter updates. As shown in Fig. 3, IPOD consistently achieves significant performance improvements over randomly initialized baselines. The error maps further validate our approach, showing reduced reconstruction errors and fewer artifacts compared to baselines.

**Convergence Evolution Analysis against SIREN Baseline**   As depicted in Fig. 4, we analyze the convergence of IPOD-initialized SIREN against randomly initialized baseline. The performance curves demonstrate that IPOD achieves significantly faster convergence and superior final performance. The reconstructed images further validate this efficiency, with IPOD-SIREN producing notably better results at early iterations (50 and 100) for both Cartesian 1D and radial 2D sampling. Error maps reveal that IPOD learns to remove reconstruction artifacts much faster, achieving superior image quality and PSNR scores by 150 iterations.

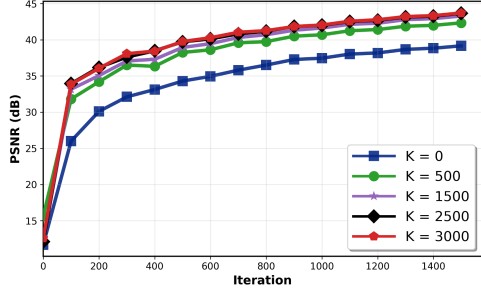

Figure 6: The performance and convergence comparison between baseline SIREN (K = 0) and IPOD-SIREN with different meta-learning training epochs (K) on unseen samples from the fastMRI brain dataset for AF = 4.

**Results for Real-World Data Reconstruction**   As shown in Fig. 5, we evaluated baselines with and without IPOD initialization on real-world prospectively undersampled data. Notably, the T1w modality was unseen during meta-learning training, yet IPOD remained effective with only 75 parameter updates. IPOD-initialized models achieve superior reconstruction quality, producing sharper images with reduced artifacts. This is particularly evident in magnified regions, where IPOD successfully recovers structures that exhibit distortions in

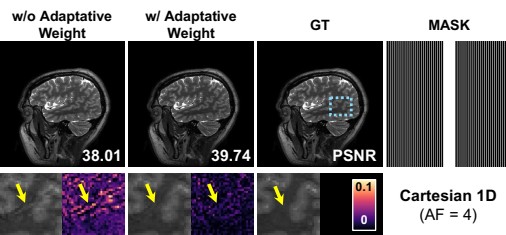

Figure 7: Qualitative and quantitative comparison showing the influence of the adaptive weight mechanism in the IPOD framework. Results are from IPOD-SIREN on a sample from the MoDL dataset.

Table 2: Quantitative evaluation showing the influence of the adaptive weight mechanism in the IPOD framework on reconstruction performance. Results are from IPOD-SIREN on the MoDL dataset for AF = 4.

| Method | PSNR (dB) | SSIM |
|---|---|---|
| w/o Adaptive Weight | 36.38±2.20 | 0.978±0.025 |
| w/ Adaptive Weight | **37.14±2.41** | 0.980±0.023 |

baseline reconstructions, validating IPOD's strong generalization capability to real-world measurements.

## 4.3 ABLATION STUDIES

**Influence of Meta-learning epochs** Fig. 6 illustrates the performance of the SIREN network as a function of meta-learning epochs (K). The curves show that even a small number of epochs (K = 500) significantly boosts the performance of our IPOD-initialized SIREN networks over its randomly initialized counterpart (K = 0). As K increases, the effectiveness of initialization steadily improves, leading to faster convergence and higher final PSNR scores before reaching a plateau. This shows that our IPOD framework enables the baseline model to learn a more robust initialization for MRI reconstruction.

**Influence of Adaptive Weight** Table 2 demonstrates the effectiveness of the adaptive weight mechanism for efficient initialization within our IPOD framework, showing a significant drop in reconstruction performance when this component is removed. This effect is further illustrated in Fig. 7, where the model trained without adaptive weights produces noticeable artifacts in the final reconstructed images. This clearly indicates that without this mechanism, the meta-learning process incorporates suboptimal parameter distributions, which negatively impacts the final reconstruction quality.

## 5 CONCLUSION

We presented **IPOD**, an inverse-problem-driven meta-learning framework that learns robust INR initialization without fully-sampled reference data. Our approach addresses the difficulties preventing existing meta-learning frameworks from direct application to medical image reconstruction through physics-informed meta-optimization and adaptive weighting mechanisms. Comprehensive experimental validation on simulated and real-world datasets demonstrates that IPOD achieves faster convergence and superior reconstruction quality across diverse imaging scenarios, including unseen subjects, contrast mechanisms, and sampling patterns, while enabling significantly faster adaptation compared to baseline INR models.

**Limitation** Despite its strong effectiveness, our framework offers room for enhancement. A key limitation is the absence of textual information embedding, which could enrich multi-modality prior knowledge by integrating imaging protocol descriptions and clinical parameters for more informed reconstruction strategies. Another limitation is our focus on 2D reconstruction tasks, while substantial potential exists for 3D extension. The volumetric structure of 3D MRI data exhibits larger shared feature similarity distributions across slices, providing stronger statistical foundations for meta-learning and potentially greater initialization effectiveness where computational efficiency gains would be most pronounced.

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

# A APPENDIX

## A.1 USE OF LLMS

We affirm that our work was conducted in accordance with the ICLR guidelines regarding the use of large language models (LLMs). LLMs were utilized solely for minor text editing, specifically for correcting spelling and grammar errors. No LLM was used to generate or alter any core technical content, including the research methodology, experimental design, results, or conclusions. The ideas, data analysis, and scientific conclusions presented in this paper are the sole product of the authors' original work.

## A.2 ADDITIONAL DETAILS OF PRE-PROCESSING

In the validation experiments, the generation of 2D Gaussian and Poisson under-sampling masks followed the methodology described in (Chung & Ye, 2022).

## A.3 ADDITIONAL DETAILS OF METRICS AND IMPLEMENTATION

**Metrics for MR Image Quality**  In our evaluation, two commonly used visual metrics—PSNR and SSIM—are employed to assess the quality of reconstructed MR images. These metrics are implemented using the Python library `skimage` (**https://github.com/scikit-image/scikit-image**).

**Implementation for Network Structures and Testing**  The network architectures for $f_{\text{real}}$ and $f_{\text{imag}}$ vary across methods. For DINER, both networks use MLPs with 2 hidden layers of 16 neurons each, employing ReLU activation functions. SIREN networks consist of MLPs with 6 hidden layers of 256 neurons each, utilizing sinusoidal activation functions as specified in the original work. For HASH, both networks combine a hash-encoding module with an MLP containing 2 hidden layers of 16 neurons each with ReLU activations. We adopt the hyperparameters suggested in the original paper (Müller et al., 2022) for the hash-encoder configuration.

In the testing-time evaluation, all methods employ the AdamW optimizer with a fixed weight decay of $2 \times 10^{-4}$. For the DINER method, the initial learning rate is set to $1 \times 10^{-2}$ with a decay factor of 0.6 every 100 steps. For the SIREN method, the initial learning rate is $1 \times 10^{-4}$, where the baseline decays by 0.8 every 500 steps while the IPOD-initialized version decays by 0.8 every 100 steps. For the HASH method, the hash encoder uses a learning rate of $5 \times 10^{-2}$ and the MLP parameters use $1 \times 10^{-3}$, with both components decaying by 0.8 every 100 steps.

## A.4 ADDITIONAL DETAILS OF BASELINES

**DINER**  Disorder-Invariant Implicit Neural Representation (DINER), a framework that largely solves this frequency-related issue by re-arranging the coordinates of the input signal (Xie et al., 2023). DINER augments a traditional INR backbone with a learnable hash-table. This innovation allows the framework to handle discrete signals that share the same attribute histogram but have different spatial arrangements. The hash-table maps the input coordinates into a consistent distribution, regardless of the original arrangement order. This transformation results in a low-frequency mapped signal that can be better modeled by the subsequent INR network, leading to a significantly alleviated spectral bias. In this work, we reproduce the MLP-based DINER structures based on their official code (**https://github.com/Ezio77/DINER**).

**SIREN**  Implicit Neural Representation is a powerful paradigm, but its effectiveness is limited by its inability to accurately model signals with fine detail and spatial/temporal derivatives. Sinusoidal Representation Network (SIREN) leverages periodic activation functions to overcome these limitations (Sitzmann et al., 2020b). SIREN is ideally suited for representing complex natural signals and their derivatives. SIREN has been proven to be a superior self-supervised INR framework for inverse problems in MRI reconstruction (Feng et al., 2023). In this work, we reproduce the SIREN structures based on their official code (**https://github.com/vsitzmann/siren**).

**HASH**  Here, HASH refers to the Instant Neural Graphics Primitives with a Multiresolution Hash Encoding (InstantNGP) method (Müller et al., 2022). The method combines lightweight MLP networks with multi-resolution hash encoding, where coordinates are mapped through hash functions to feature vectors at different spatial resolutions. Multi-resolution hash encoding replaces traditional positional encoding with efficient hash tables at multiple resolution levels, significantly reducing computational overhead while maintaining high-quality representations. In this work, we reproduce the HASH structures based on their official code (**https://github.com/NVlabs/instant-ngp**).

## A.5 ADDITIONAL VISUAL RESULTS

Fig. 8, Fig. 9, Fig. 10, and Fig. 11 show additional reconstructed MR images through different baseline methods with and without IPOD initialization. The proposed IPOD initialization achieves fast adaptation across various imaging modalities and different inverse problems in MRI reconstruction.

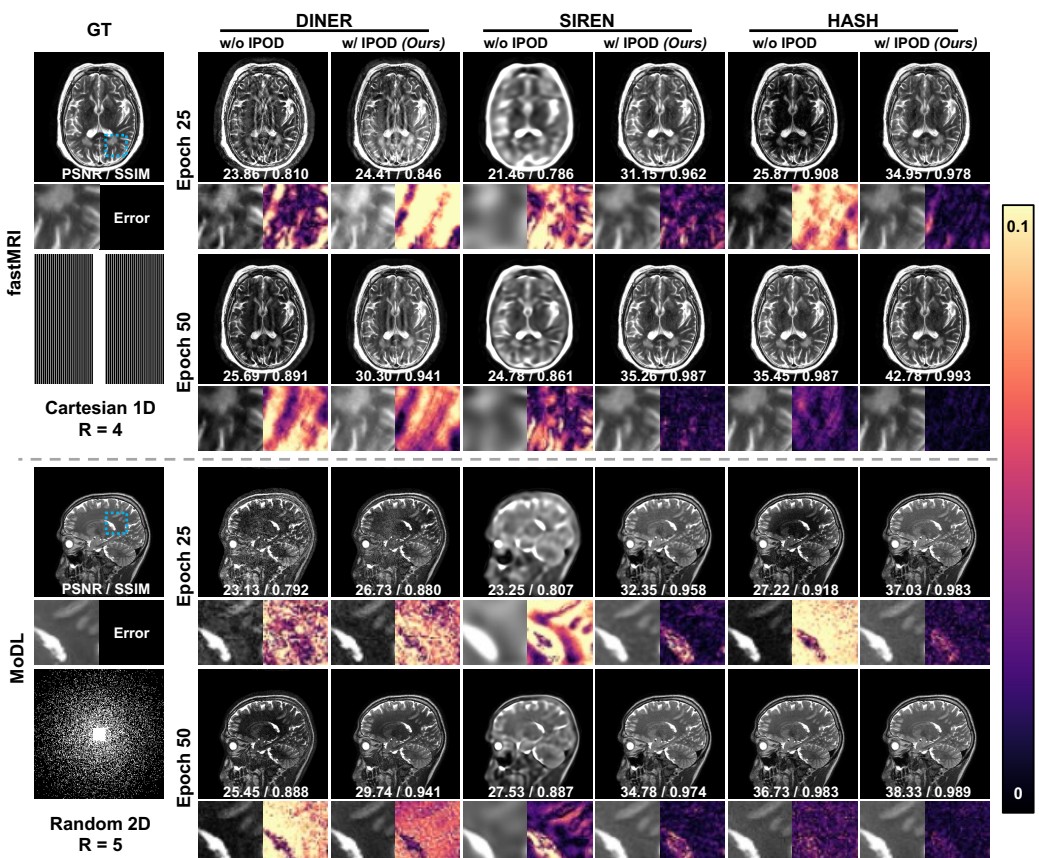

Figure 8: Qualitative and quantitative comparison of baselines with and without IPOD initialization under different parameter updates on fastMRI and MoDL datasets.

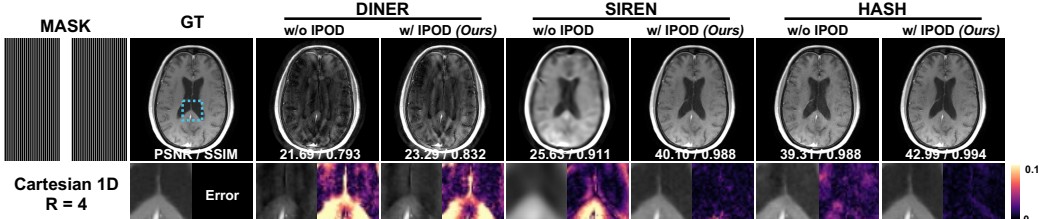

Figure 9: Qualitative and quantitative comparison of baselines with and without IPOD initialization under a small number of parameter updates (75 iterations) on the unseen T1w modality in fastMRI for Af = 4.

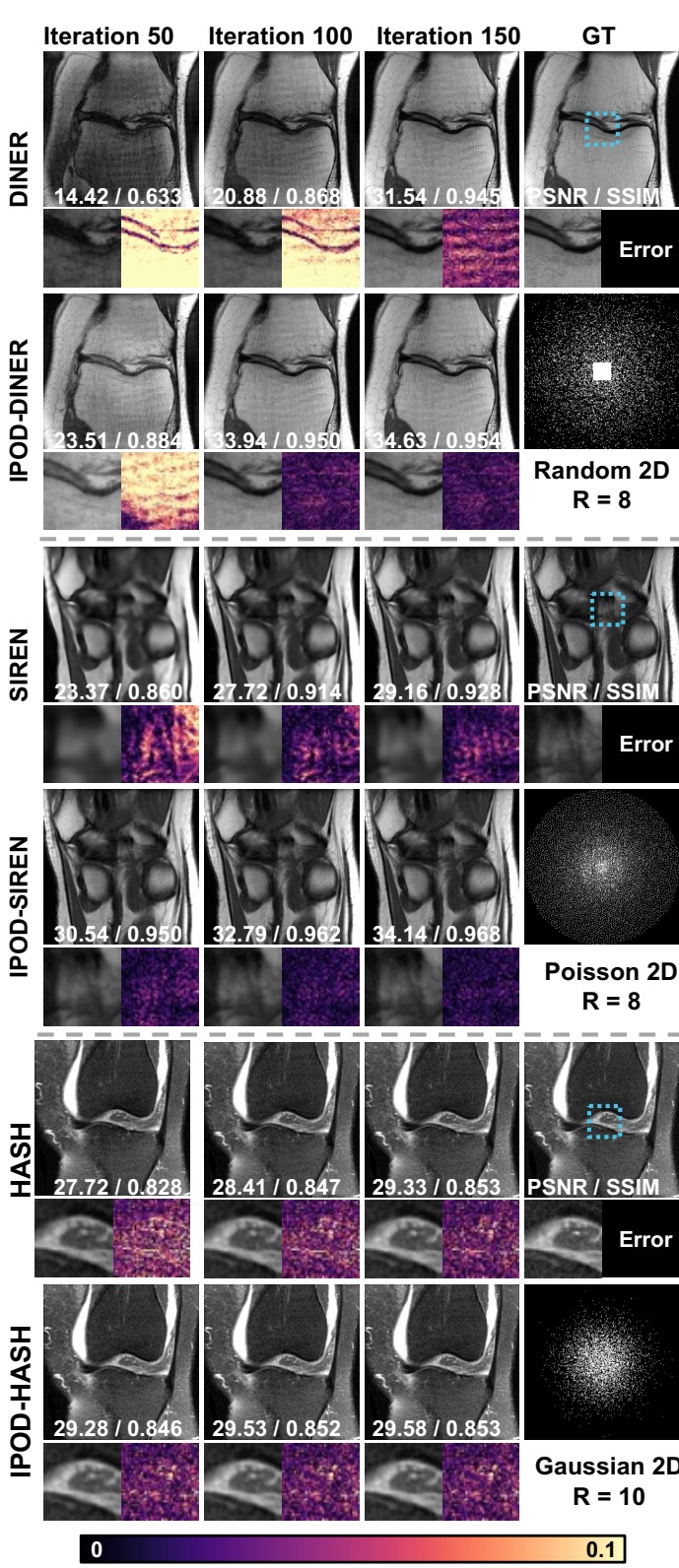

Figure 10: Qualitative and quantitative comparison of baselines with and without IPOD initialization under different parameter updates on fastMRI Knee datasets with different mask patterns.

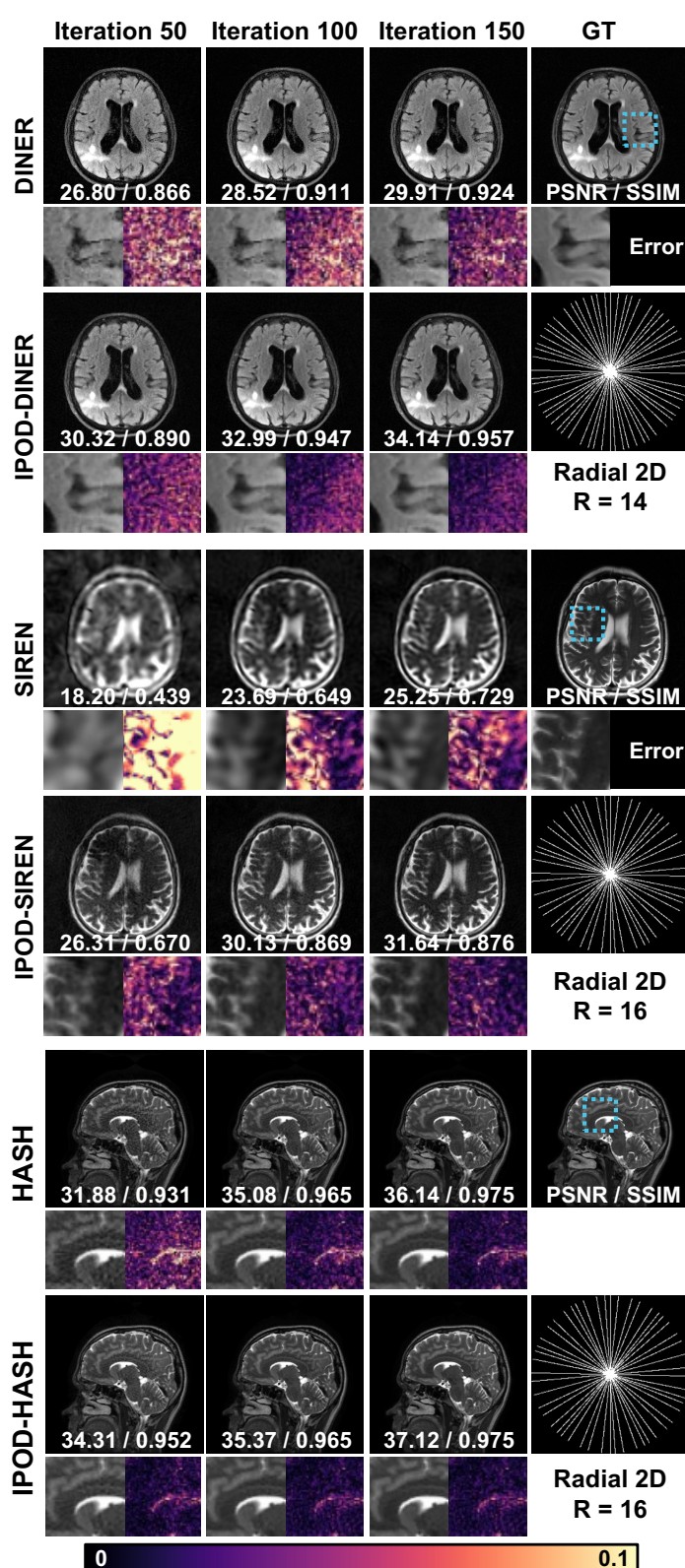

Figure 11: Qualitative and quantitative comparison of baselines with and without IPOD initialization under different parameter updates on fastMRI and MoDL datasets with different radial sampling patterns.

