# SUPPLEMENTARY

## 1 ABLATION STUDY OF DIVERSITY IN PROBLEM SET (REVIEWER MI8R W2)

To evaluate the effect of problem set diversity, we have conducted the exact ablation study requested to demonstrate that the diverse problem set provides robust priors. In this ablation study, we compared three meta-learning strategies:

1. **IPOD-Brain**: The problem set in the IPOD learning phase specialized only on brain anatomy data.

2. **IPOD-Cartesian**: The problem set in the IPOD learning phase specialized only on data from Cartesian undersampling.

3. **IPOD-All**: The whole problem set (multiple anatomies and undersampling patterns) used in the IPOD learning phase.

Note that all the hyperparameters in these three strategies remained consistent with those reported in the main paper. We compare the prior generalization of IPOD-Cartesian and IPOD-All on the radial undersampled dataset (shown in Figure S1 and Table S1), while comparing IPOD-Brain and IPOD-All on knee data (shown in Figure S2 and Table S2).

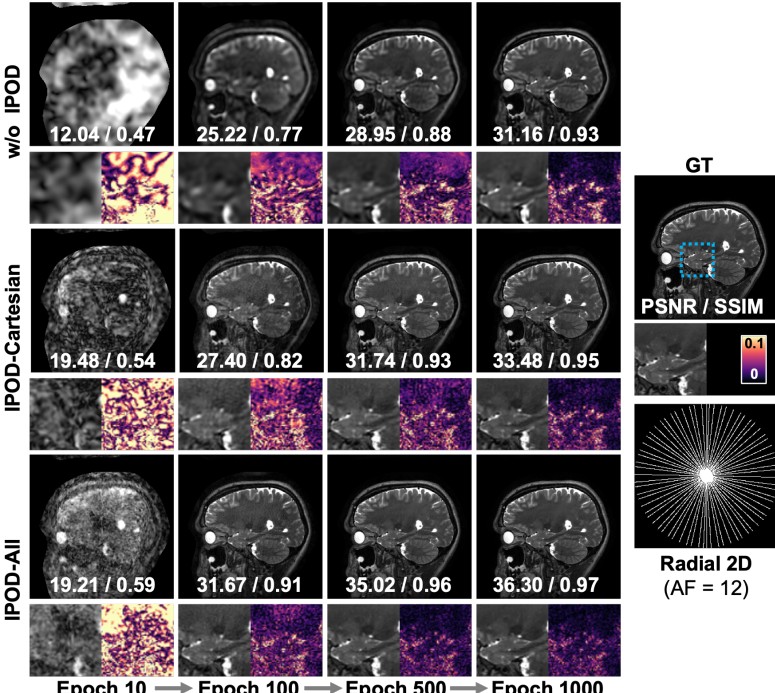

Figure S1: Qualitative and quantitative comparison of Baselines and IPOD with different meta-learning strategies across progressive iterations on MoDL dataset. (IPOD-Cartesian: meta-trained on Cartesian undersampling pattern only; IPOD-All: meta-trained on diverse undersampling patterns.)

Table S1: Quantitative comparison (Mean ± Std in PSNR/SSIM) of Baselines and IPOD with different meta-learning strategies across progressive iterations on the radial undersampling pattern. (IPOD-Cartesian: meta-trained on Cartesian undersampling pattern only; IPOD-All: meta-trained on diverse undersampling patterns.)

| Method | Iter 10 | | Iter 100 | | Iter 500 | | Iter 1000 | |
|---|---|---|---|---|---|---|---|---|
| | PSNR | SSIM | PSNR | SSIM | PSNR | SSIM | PSNR | SSIM |
| w/o IPOD | 12.74±0.98 | 0.510±0.056 | 23.90±1.87 | 0.772±0.006 | 29.17±0.31 | 0.891±0.017 | 31.47±0.45 | 0.934±0.006 |
| IPOD-Cartesian | 16.40±4.36 | 0.530±0.017 | 26.80±0.84 | 0.832±0.011 | 31.91±0.25 | 0.927±0.002 | 32.92±0.79 | 0.953±0.008 |
| IPOD-All | **18.51±1.00** | **0.604±0.022** | **30.97±0.99** | **0.907±0.009** | **33.78±1.76** | **0.957±0.007** | **34.21±2.96** | **0.968±0.006** |

Table S1 and S2 demonstrate that IPOD-All achieves the best performance in all tests with faster convergence, while Figure S1 and S2 show that IPOD-Brain and IPOD-Cartesian at the early iteration stage (100 iterations) present artifacts and blurring of structural details, which are not observed in IPOD-All. Furthermore, all these three strategies offer a prior that accelerates the network adaptation to the unseen data compared with that without initialization from IPOD.

The fact that IPOD-All outperforms IPOD-Brain on knee data and outperforms IPOD-Cartesian on radial sampled data directly proves our central claim: exposure to diverse inverse problems during IPOD meta-learning prevents prior overfitting to specific problem characteristics and produces more robust, generalizable initialization.

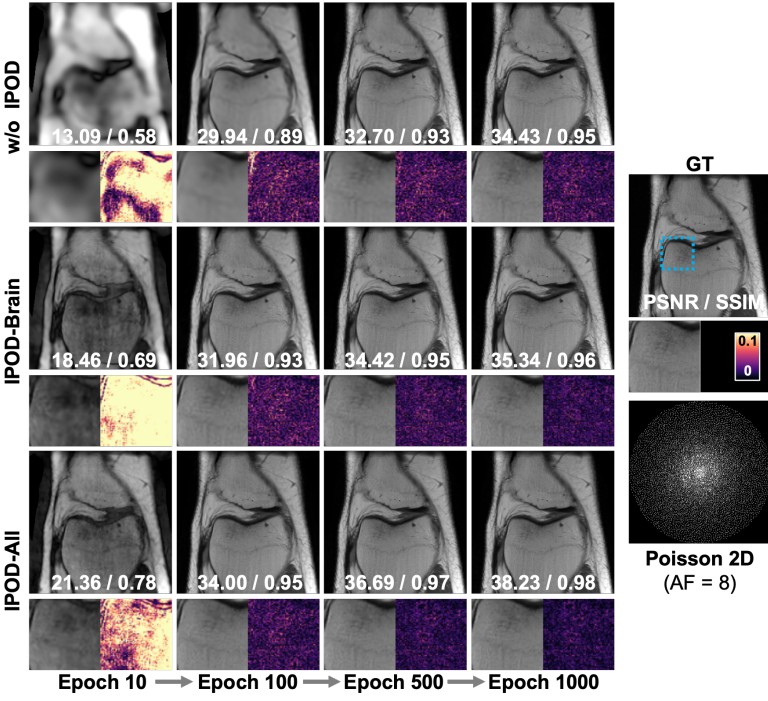

Figure S2: Qualitative and quantitative comparison of Baselines and IPOD with different meta-learning strategies across progressive iterations on fastMRI knee dataset. (IPOD-Brain: meta-trained on brain dataset only; IPOD-All: meta-trained on diverse anatomies dataset.)

## 2 ADDITIONAL COMPARISON EXPERIMENTS (REVIEWER 26VT Q1 AND Q2)

We added comprehensive comparison experiments at multiple acceleration factors with different methods. To thoroughly evaluate IPOD's performance, we compared against hand-crafted regularization techniques, such as total variation (TV) and L1Wavelet, as well as the self-supervised method SSDU. All experiments used identical coil sensitivity maps pre-computed from the ESPIRiT

Table S2: Quantitative comparison (Mean ± Std in PSNR/SSIM) of Baselines and IPOD with different meta-learning strategies across progressive iterations on fastMRI knee dataset. (IPOD-Brain: meta-trained on brain dataset only; IPOD-All: meta-trained on diverse anatomies dataset.)

| Method | Iter 10 | | Iter 100 | | Iter 500 | | Iter 1000 | |
|---|---|---|---|---|---|---|---|---|
| | PSNR | SSIM | PSNR | SSIM | PSNR | SSIM | PSNR | SSIM |
| w/o IPOD | 13.81±2.62 | 0.539±0.057 | 26.59±3.11 | 0.833±0.050 | 29.99±2.81 | 0.904±0.026 | 31.71±2.83 | 0.929±0.018 |
| IPOD-Brain | 17.74±1.08 | 0.667±0.042 | 29.33±3.53 | 0.899±0.040 | 31.91±3.68 | 0.937±0.028 | 33.14±3.38 | 0.950±0.023 |
| IPOD-All | **19.57±2.58** | **0.726±0.044** | **30.72±3.35** | **0.919±0.029** | **33.04±3.93** | **0.951±0.020** | **34.74±3.58** | **0.960±0.017** |

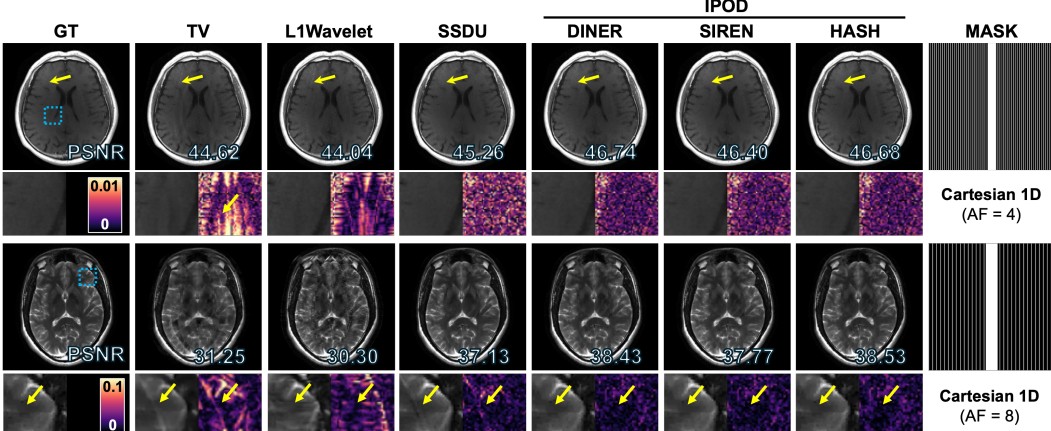

Figure S3: Qualitative and quantitative comparison of hand-crafted regularization techniques, SSDU and IPOD (with different network backbones) for 1D Cartesian undersampling.

Table S3: Quantitative comparison (Mean ± Std in PSNR/SSIM) of different reconstruction methods on 1D Cartesian undersampling pattern at different acceleration factors.

| Sampling | AF | Metric | TV | L1Wavelet | SSDU | DINER-IPOD | SIREN-IPOD | HASH-IPOD |
|---|---|---|---|---|---|---|---|---|
| 1D Cartesian | 4 | PSNR | 43.54±3.12 | 44.66±0.73 | 45.33±0.56 | **46.81±1.14** | 46.55±0.02 | 46.66±0.02 |
| | | SSIM | 0.983±0.016 | 0.988±0.008 | 0.993±0.005 | 0.997±0.002 | 0.998±0.009 | **0.998±0.007** |
| | 8 | PSNR | 27.36±3.90 | 29.19±1.11 | 35.72±1.61 | 36.66±2.51 | **36.81±1.37** | 36.68±2.62 |
| | | SSIM | 0.927±0.013 | 0.928±0.006 | 0.938±0.011 | 0.943±0.005 | 0.940±0.001 | **0.947±0.005** |

method. Especially for SSDU, we adhered to the protocol in (Yaman et al., 2020) with identical network architectures and hyperparameters. The training dataset matches that used in IPOD meta-learning. Additionally, for a fair comparison, the pre-trained SSDU model was carefully fine-tuned for 500 iterations on test data to ensure optimal performance.

As shown in Table S3, all IPOD-based methods achieve superior reconstruction quality compared to hand-crafted and self-supervised baselines in both PSNR and SSIM metrics for both acceleration factors. Meanwhile, Figure S3 shows that IPOD-based methods effectively suppress aliasing artifacts in challenging anatomical regions (yellow arrows in zoom-in panels), which remain prominent in TV, L1Wavelet, and SSDU reconstructions.

These results confirm that IPOD remains effective in 1D Cartesian undersampling with different acceleration factors, compared to hand-crafted regularization techniques and SSDU.

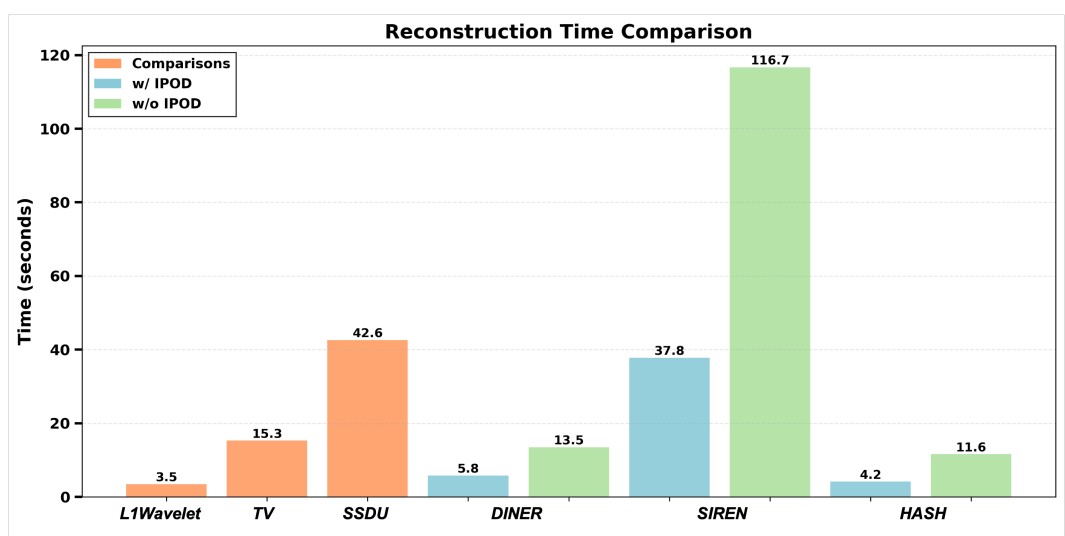

Figure S4: Qualitative and quantitative comparison of hand-crafted regularization techniques, SSDU and IPOD (with different network backbones) across Cartesian, Gaussian, and Poisson undersampling patterns.

## 3 RECONSTRUCTION TIME CONSUMPTION COMPARISONS (REVIEWER 26VT Q3)

To evaluate the computational efficiency of IPOD, we conducted comprehensive runtime comparisons across different reconstruction methods, including hand-crafted regularization techniques, self-supervised methods, and INR-based approaches with and without IPOD initialization.

All experiments were conducted on an NVIDIA A100 GPU with image size 256×256 per 2D slice. We employed a consistent convergence criterion (PSNR stabilization with change rate $< 0.01$ dB/iteration) to ensure all methods reach equivalent convergence quality. Reported times represent pure reconstruction time (excluding data I/O), averaged across Cartesian (AF=4), Gaussian (AF=6), and Poisson (AF=8) undersampling patterns.

Figure S4 shows the time consumption comparison. The results show that IPOD significantly accelerates reconstruction speed across all INR backbones. Notably, HASH-IPOD (4.2s) matches the approximate speed of hand-crafted regularization methods (L1Wavelet: 3.5s).

Table S4: Reconstruction time comparison (seconds per slice) across different methods. All timings averaged across Cartesian, Gaussian, and Poisson undersampling patterns.

| Method | w/o IPOD | w/ IPOD |
|---|---|---|
| L1Wavelet | 3.5 | \ |
| TV | 15.3 | \ |
| SSDU | 42.6 | \ |
| DINER | 13.5 | 5.8 |
| SIREN | 116.7 | 37.8 |
| HASH | 11.6 | 4.2 |

## 4 CONVERGENCE COMPARISONS ON DIFFERENT FORWARD MODELS(REVIEWER J28C Q1)

As shown in Figure S5, we evaluated convergence behavior across different datasets and anatomical regions, training all networks until stabilization of the PSNR was reached, and the convergence

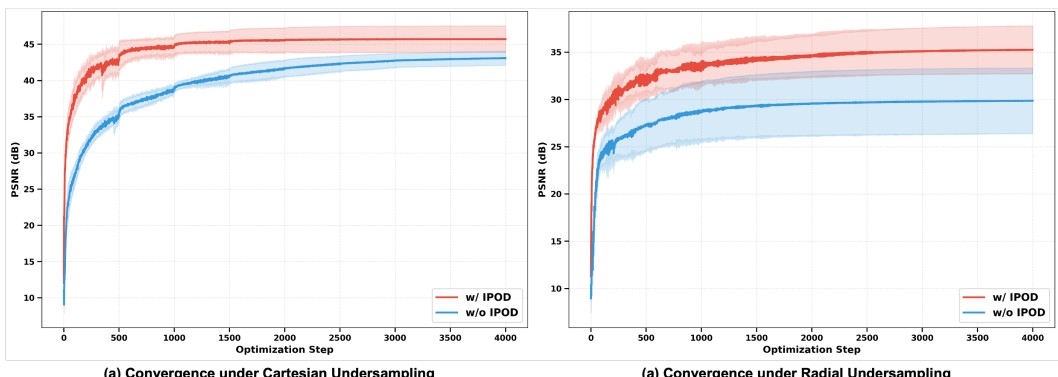

Figure S5: Comparison of performance curves of networks with and without IPOD initialization under varying sampling patterns on fastMRI and MoDL dataset.

curves demonstrate that IPOD consistently achieves a significantly higher final PSNR while simultaneously accelerating the convergence speed in all scenarios tested.

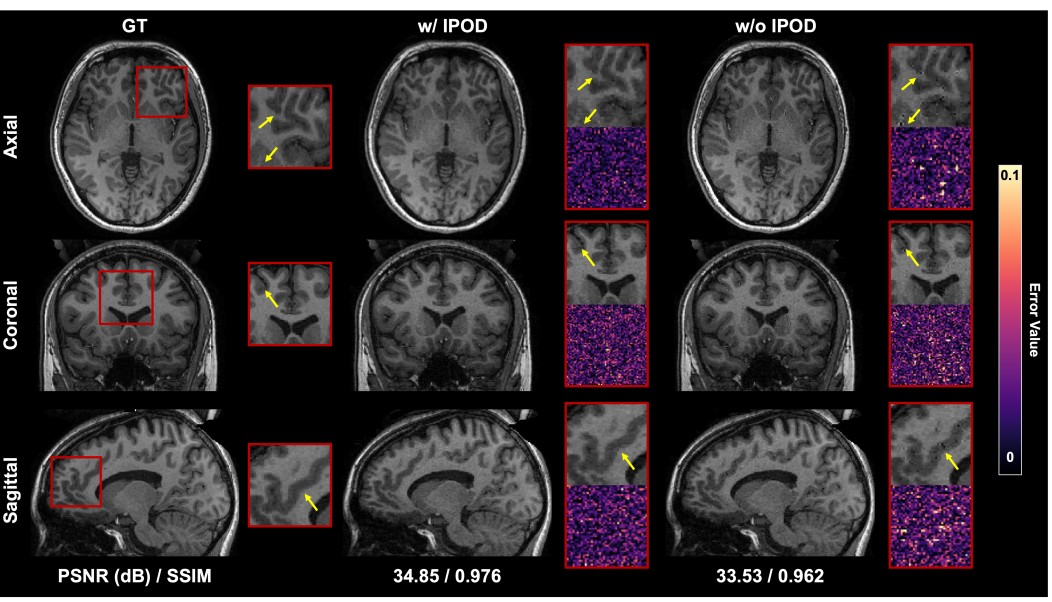

Figure S6: Qualitative and quantitative comparison of 3D reconstruction results with and without IPOD under Poisson undersampling (AF = 8). Yellow arrows indicate the abnormal structures in results without IPOD.

## 5 RECONSRTUCTION ON 3D MRI DATA(REVIEWER MI8R Q1)

To evaluate the effectiveness of our IPOD on 3D MRI data, we collected 3D brain scans from five subjects. Data were acquired using a 3D Magnetization Prepared Rapid Gradient Echo (MPRAGE) sequence with the following parameters: matrix size = $240 \times 240 \times 160$, voxel size = $1 \times 1 \times 1$ mm$^3$, TR/TE/TI = $2100/3.1/750$ ms, flip angle = $10°$, bandwidth = $250$ Hz/pixel. The raw data for $k$-space were 1D inverse Fourier transformed along the fully sampled slice direction. Each slice of $k$-space data is retrospectively undersampled with different mask patterns and acceleration factors.

As shown in Fig S6, the results demonstrate that benefiting from the learned robust prior, IPOD-reconstructed images preserve superior structural details and effectively suppress noise interference, which is particularly severe in high-resolution acquisitions. These experiments validate that IPOD's

slice-wise framework not only works in principle for 3D MRI but also delivers superior reconstruction quality in practical high-resolution 3D scenarios. The comprehensively quantitative comparison is shown in Table S5.

Table S5: Quantitative comparison (Mean ± Std in PSNR/SSIM) of 3D reconstruction results with and without IPOD across different acceleration factors and sampling patterns.

| Sampling Pattern | AF | Metric | w/ IPOD | w/o IPOD |
|---|---|---|---|---|
| 1D Cartesian | 4 | PSNR | **35.21 ± 2.09** | 34.33 ± 1.96 |
| | | SSIM | **0.985 ± 0.008** | 0.971 ± 0.009 |
| 2D Poisson | 6 | PSNR | **33.21 ± 1.37** | 31.52 ± 2.09 |
| | | SSIM | **0.978 ± 0.007** | 0.964 ± 0.009 |
| 2D Gaussian | 8 | PSNR | **31.60 ± 1.49** | 29.78 ± 1.33 |
| | | SSIM | **0.9626 ± 0.014** | 0.9555 ± 0.013 |

# 6 HYPERPARAMETER SWEEP FOR THE VANILLA INR NETWORKS(REVIEWER J28C Q2)

To guaranty the best performance of the vanilla INR networks, we conducted the learning rate sweep. Table S6 presents our comprehensive hyperparameter sweep for models without IPOD initialization across multiple orders of magnitude. The results demonstrate that our parameter settings achieve optimal performance for each architecture.

Table S6: Learning rate hyperparameter sweep for INRs without IPOD. Bold indicates optimal configuration (LR: learning rate).

| | | | | | |
|---|---|---|---|---|---|
| DINER | LR | 5.0e-02 | **1.0e-02** | 5.0e-03 | 2.0e-03 |
| | PSNR | $46.62 \pm 0.16$ | $\mathbf{46.66 \pm 0.14}$ | $46.59 \pm 0.14$ | $42.76 \pm 3.66$ |
| SIREN | LR | 2.0e-04 | **1.0e-04** | 8.0e-05 | 5.0e-05 |
| | PSNR | $40.23 \pm 1.37$ | $\mathbf{41.42 \pm 1.26}$ | $39.55 \pm 0.09$ | $35.62 \pm 1.22$ |
| HASH (Fixed MLP) | LR | 7.5e-02 | **5.0e-02** | 5.0e-03 | 1.0e-03 |
| | PSNR | $46.35 \pm 0.01$ | $\mathbf{46.63 \pm 0.15}$ | $46.49 \pm 0.04$ | $45.92 \pm 0.03$ |
| HASH (Fixed Encoder) | LR | 5.0e-03 | **1.0e-03** | 5.0e-04 | 1.0e-04 |
| | PSNR | $46.56 \pm 0.13$ | $\mathbf{46.63 \pm 0.15}$ | $46.50 \pm 0.09$ | $46.03 \pm 0.06$ |

## REFERENCES

Burhaneddin Yaman, Seyed Amir Hossein Hosseini, Steen Moeller, Jutta Ellermann, Kâmil Uğurbil, and Mehmet Akçakaya. Self-supervised learning of physics-guided reconstruction neural networks without fully sampled reference data. *Magnetic resonance in medicine*, 84(6):3172–3191, 2020.