# OpenReview forum: "IPOD:Inverse-Problem-Driven Meta-Learning for Fast Generalizable Neural Representations in MRI Reconstruction"
_ICLR.cc/2026/Conference — ICLR 2026 Conference Withdrawn Submission_

### Official Review · Reviewer_j28c · 2025-10-27

**Soundness:** 3
**Presentation:** 2
**Contribution:** 2
**Rating:** 2
**Confidence:** 4

**Summary:**

This paper introduces IPoD, a meta-learning framework for implicit neural representations, to accelerate the reconstruction and super-resolution of undersampled MRI. For this, the authors utilize an established meta-learning framework based on Tancik et al., and integrate the forward k-space physical model, typically used in k-space INRs, into the meta-learning process. Leveraging the proposed framework, the authors are able to outperform subject-specific MRI baselines trained on individual scans only by high margins.

**Strengths:**

- The paper is well-written, has strongly motivated reasoning as to why meta-learning could be a beneficial candidate for accelerating k-space INRs, and provides convincing results that this enables accelerated learning.
- The introduction of the physical model in the methods section reads particularly well, and is important because it introduces the reader to a concept that is at the core of this paper.
- In experiments, the proposed method is able to outperform the baseline methods by high margins even though a perceptual metric such as LPIPS would have helped to assess the reconstruction quality even more.

**Weaknesses:**

- Novelty of the proposed method:

While the results are encouraging and convincing for the presented method, the work lacks technical novelty. It builds upon the framework introduced in [9], which has already shown that meta-learning single-instance INRs is beneficial (also in the context of CT / medical images), and merely adds the physical forward model to conduct this in the k-space, which is not exactly new since INRs have been used in k-space before as well, e.g. in [10]. In conclusion, I believe this paper does not propose the novelty expected for ICLR and would be better suited for an application-centered presentation e.g. at a medical conference such as MIDL or MICCAI, or a medical journal paper (e.g., TMI, MEDIA). However, I would strongly encourage this, given the merit of the work and the value it may provide to the k-space reconstruction community.

The authors state that "For medical image reconstruction inverse problems, the potential of meta-learning-based INR initialization remains unexplored." I would argue that this has been studied in a number of publications (e.g., [4, 5, 6, 8]), and some of these, while possibly still under review, have been released on arXiv at the beginning of this year, or published, eg at MICCAI last year [6].

- Structure of Introduction and Related Work:

While the paper subtly touches upon some of these concepts, I feel it would benefit from a clearer structure, especially in the Related Work section. The paper lacks a discussion and contextualization of the work within the field of INR approaches in medical imaging, especially regarding the "physical modeling prior." It would be beneficial to know if related works use similar physical priors (I suspect they do, as this is standard), but the absence of this discussion is problematic since it is the main contribution of the paper.

Also regarding architectures, the INR medical imaging community typically distinguishes between instance-specific (single-subject) INRs (which the authors use) and cohort-learned INRs, where multiple images are modeled by the same network (e.g., [4]). The authors should be aware of this distinction since they cite [3, 5]. For cohort-learned INRs, different architectures exist, most of which use a modulation-based approach [1, 2, 3]. I feel that a reader who is not aware of this distinction won't be able to properly place the work within the context of current medical INR works. Moreover, the authors do not state why the single-instance framework may be of particular relevance in this case, since cohort-learned INRs may provide even higher incentive given that they model a dataset jointly [4,7]. At the very least, they should acknowledge this limitation.

- Minor: There is a typo in line 052 (as indicated in the original PDF, though not visible here).

[1] Park, Jeong Joon, et al. "Deepsdf: Learning continuous signed distance functions for shape representation." Proceedings of the IEEE/CVF conference on computer vision and pattern recognition. 2019.

[2] Mehta I, Gharbi M, Barnes C, Shechtman E, Ramamoorthi R, Chandraker M. Modulated periodic activations for generalizable local functional representations. InProceedings of the IEEE/CVF International Conference on Computer Vision 2021 (pp. 14214-14223).

[3] Dupont, Emilien, et al. "From data to functa: Your data point is a function and you can treat it like one." arXiv preprint arXiv:2201.12204 (2022).

[4] Dannecker M, Kyriakopoulou V, Cordero-Grande L, Price AN, Hajnal JV, Rueckert D. CINA: conditional implicit neural atlas for spatio-temporal representation of fetal brains. InInternational Conference on Medical Image Computing and Computer-Assisted Intervention 2024 Oct 3 (pp. 181-191). Cham: Springer Nature Switzerland.

[5] Bauer M, Dupont E, Brock A, Rosenbaum D, Schwarz JR, Kim H. Spatial functa: Scaling functa to imagenet classification and generation. arXiv preprint arXiv:2302.03130. 2023 Feb 6.
[6] De Paolis GR, Lenis D, Novotny J, Wimmer M, Berg A, Neubauer T, Matthias Winter P, Major D, Muthusami A, Schröcker G, Mienkina M. MICCAI ShapeMI Workshop.

[7] Friedrich P, Bieder F, Cattin PC. MedFuncta: Modality-Agnostic Representations Based on Efficient Neural Fields. arXiv preprint arXiv:2502.14401. 2025 Feb 20.

[8] Dannecker M, Sanchez T, Cuadra MB, Turgut Ö, Price AN, Cordero-Grande L, Kyriakopoulou V, Hajnal JV, Rueckert D. Meta-learning Slice-to-Volume Reconstruction in Fetal Brain MRI using Implicit Neural Representations. arXiv preprint arXiv:2505.09565. 2025 May 14.

[9] Tancik M, Mildenhall B, Wang T, Schmidt D, Srinivasan PP, Barron JT, Ng R. Learned initializations for optimizing coordinate-based neural representations. InProceedings of the IEEE/CVF conference on computer vision and pattern recognition 2021 (pp. 2846-2855).

[10] Huang W, Li HB, Pan J, Cruz G, Rueckert D, Hammernik K. Neural implicit k-space for binning-free non-cartesian cardiac MR imaging. InInternational Conference on Information Processing in Medical Imaging 2023 Jun 8 (pp. 548-560). Cham: Springer Nature Switzerland.

**Questions:**

1. PSNR Improvement: What is the authors' intuition for attaining much higher PSNR scores? From what I have read in other meta-learning papers, the benefit of using meta-learning typically lies in the convergence speed, and not necessarily in an improvement of the converged networks. Have all networks converged in your experiments?

2. Hyperparameter Sweep: Did you conduct a learning rate hyperparameter sweep for the non-meta-learned INRs? Since meta-learning heavily changes the training dynamics, it would be important to report the best learning rate configuration for the vanilla models, especially since the meta-learning framework has its own optimization loop.

3. "Unsupervised" Paradigm: In several sections of the paper (Abstract, Introduction, Methods), the authors present INRs as an "unsupervised paradigm." Could the authors please elaborate on this? I personally believe they are quite the contrary: they overfit to signals, they embed information in their weights (or latents in the context of modulated INRs), and they have a supervised loss (even in the context of k-space INRs).

4. Transferability Claim: In lines 098-100, the authors state: "This limitation is a primary reason for their unstable performance and relatively slow reconstruction speeds, as each new dataset requires training from scratch, and the learned representations are difficult to transfer even to similar data domains." Is there any evidence to support the claim that meta-learning makes this process more stable? Why would meta-learning avoid this, and do your experiments validate this claim?

---

> ### Author Response · Authors · 2025-11-22
>
> We sincerely appreciate your time and effort in reviewing our work.
>
> # Clarifications of Two Key Misunderstandings
>
> First, we would like to clarify two key points that may help resolve possible misunderstandings regarding our contributions:
>
> - **Our work focuses solely on MRI reconstruction, NOT super-resolution.** MRI reconstruction aims to recover high-quality MR images from undersampled k-space data, whereas super-resolution seeks to obtain high-resolution images from low-resolution inputs. The former is a pre-acquisition reconstruction task, while the latter is a post-processing task. These two problems are fundamentally different in their formulations.
>
> - **Our INR represents image-domain signals, NOT k-space directly.** NIK [10] learns a mapping from coordinates to k-space, whereas our INR maps coordinates to complex-valued MR images and uses the MRI forward model (i.e., Fourier transform) to compute the k-space loss. These represent fundamentally different learning paradigms.
>
> ---
>
> **W1.1: Novelty of the proposed method compared to Ref [9]**
>
> We respectfully but strongly disagree with this assessment, which fundamentally misrepresents our contributions.
>
> **A1:** Our work goes substantially beyond [9]. While [9] demonstrated meta-learning can accelerate INR convergence for visual tasks, it has many critical differences with our IPOD:
>
> - **Problem differences**: Ref [9] focuses on INR initialization via meta-learning for visual tasks (e.g., image regression, novel view synthesis). Although Ref [9] includes a preliminary CT reconstruction experiment, it is limited to phantom data. In contrast, MRI reconstruction is fundamentally different (complex-valued computations, diverse anatomies, multi-center variability). Therefore, applying Ref [9] to MRI reconstruction is **NOT** trivial.
>
> - **Architecture generalization:** [9] did not study effectiveness across different INR architectures. We systematically evaluate three backbones, establishing architecture-specific trade-offs applicable to inverse problems broadly.
>
> - **Diversity-robustness relationship:** [9] did not study how training set diversity affects prior quality. Our ablation studies (Figures S1-S2, Tables S1-S2) prove diverse problem sets create more robust priors, establishing design principles for meta-learning in inverse problems.
>
>
> **W1.2: Novelty of the proposed method compared to Ref [10]**
>
> **A2:** The statement "INRs have been used in k-space before [10]" reflects a fundamental misunderstanding. The work [10] directly predicts k-space values, while IPOD uses INR to represent images with forward model enabling self-supervised learning. These are entirely different paradigms. Our work has no relation to k-space-based INR.
>
> **W1.3: Novelty of the proposed method compared to Ref [4,6,7,8]**
>
> **A3:** First, the cited works [4,6,7,8] mainly aim to recover high-resolution volumes from low-resolution images, which are image post-processing tasks. This is fundamentally different from the undersampled MRI reconstruction our work focused on. Secondly, due to lack of physical models, their frameworks cannot handle undersampled data. Additionally, works [4,6,7] still depend on fully-sampled data for meta-learning, which is not required in our IPOD.
>
> The assessment appears based on misunderstandings (treating our framework as "k-space INR," comparing to representation learning works addressing different problems). We respectfully but strongly maintain that IPOD presents methodological contributions appropriate for ICLR—a reference-free meta-learning framework for inverse problem reconstruction from undersampled raw data, with broad applicability across computational imaging domains.
>
> ---

---

> ### Author Response · Authors · 2025-11-22
>
> **W2.1: Motivation for incorporating the physical forward model**
>
> **A1:** The reviewer suggests that integrating the physical forward model is our main contribution. We clarify that this is not the case. The motivation for incorporating the physical forward model is to enable this reference-free meta-learning paradigm. Therefore, the key innovation is not the forward model itself but enabling meta-learning to work in this self-supervised regime, fundamentally changing the data requirements for meta-training. Moreover, through meta-training on a diverse problem set—another key innovation of our work—IPOD generalizes effectively across varying MRI reconstruction scenarios (different sampling patterns, anatomies, and acceleration factors).
>
> **W2.2: Differences between cohort-learned INRs and our IPOD**
>
> **A2:** Existing cohort-learned INRs, such as [4] CINA (Dannecker et al., 2024) and [7] MedFuncta (Friedrich et al., 2025), are approaches mainly for representation learning and operate as post-processing tasks depending on high-quality reference images. Furthermore, cohort-learned INRs typically extract latent codes from reference images to modulate network weights. However, our IPOD addresses inverse problems from undersampled raw data. Due to the lack of references, cohort-learned INRs are not suitable to solve MRI reconstruction problems. In summary, existing cohort-learned INRs and our IPOD are fundamentally different in terms of both problem formulation and data requirements. We will include this discussion in the revised manuscript.

---

> ### Author Response · Authors · 2025-11-26
>
> **Q1: PSNR Improvement: What is the authors' intuition for attaining much higher PSNR scores? From what I have read in other meta-learning papers, the benefit of using meta-learning typically lies in the convergence speed, and not necessarily in an improvement of the converged networks. Have all networks converged in your experiments?**
>
> **A:**
> - **All networks in our experiments have fully converged.** As shown in `Figure S5 in supplementary`, we trained  networks until convergence plateaus were reached. The convergence curves demonstrate that IPOD not only achieves significantly higher final PSNR but also accelerates convergence speed.
>
> - **Meta-learning improving final quality is not unique to our work.** Recent studies [1-4] have consistently observed that meta-learning is capable of improving INRs' final performance (including PSNR) across diverse domains including image fitting, novel view synthesis, volume encoding, and video generation. These works demonstrate that meta-learning promotes solution quality in INR-based tasks, not merely training speed.
>
> - **Why this improvement also occurs  in MRI reconstruction:** MRI reconstruction from undersampled k-space is an ill-posed inverse problem with multiple local minima of varying quality. Random initialization often converges to suboptimal solutions with aliasing artifacts. Meta-learned initialization places the network in a more optimal solution space, resulting in superior final performance.
>
> > [1] Chen, Yinbo, and Xiaolong Wang. "Transformers as meta-learners for implicit neural representations." European Conference on Computer Vision. Cham: Springer Nature Switzerland, 2022.
> >
>
> > [2] Vyas, Kushal, et al. "Learning transferable features for implicit neural representations." Advances in Neural Information Processing Systems 37 (2024): 42268-42291.
> >
>
> >[3] Yang, Maizhe, Kaiyuan Tang, and Chaoli Wang. "Meta-INR: Efficient Encoding of Volumetric Data via Meta-Learning Implicit Neural Representation." 2025 IEEE 18th Pacific Visualization Conference (PacificVis). IEEE, 2025.
> >
>
> >[4] Guo, Jialong, et al. "MetaNeRV: Meta Neural Representations for Videos with Spatial-Temporal Guidance." Proceedings of the AAAI Conference on Artificial Intelligence. Vol. 39. No. 3. 2025.

---

> ### Comment · Reviewer_j28c · 2025-11-27
> **Response to Rebuttal - Part 1**
>
> Dear Authors,
>
> I thank you for the detailed rebuttal and the clarifications regarding the technical distinctions between your work and other MRI reconstruction literature. I also greatly appreciate your reply regarding the intuition for attaining much higher PSNR scores in your application.
>
> ### Acknowledgment of oversight regarding technical distinctions in my summary, request to position the paper more clearly
>
> I fully acknowledge and apologize for the oversight in my initial review regarding the specific training scheme. I appreciate the authors' clarification that IPOD operates in the image domain (mapping coordinates to complex-valued images) and uses the physical forward model (similar to Feng et al., and as reference by the authors in l.154). I agree, that this is inherently different from k-space INRs (e.g. Spieker et al. [2,3]) that model k-space directly in the INR. I also acknowledge that I blurred the lines between the "reconstruction" task and the "super-resolution" tasks, which may have deterred from the main message of my review and concerns.
>
> However, I also believe that discrediting my valid concerns regarding both, technical novelty and paper presentation, over wording/misinterpretation in my summary of the paper, is not warranted. In fact, as I understand the ICLR process, it is entirely encouraged to change the presentation of the paper, e.g. re-write parts of the Intro and Related Work to reflect new references and feedback from the reviewers, aand to discuss misunderstandings. That being said, I would like to briefly re-iterate my points:
>
> ### Clarity of Presentation and Positioning of the paper
>
> (i) My confusion regarding the method's positioning, which you noted in the rebuttal, partially stems from your manuscript's structure..
> The authors cite a vast corpus of INR MRI recon literature in the first paragraph, but don't discuss the difference of proposed methods. I agree that I falsely binned their work into k-space recon INRs, instead of image-space recon INRs, but it also not properly delineated. Please revise your Related Work section to reflect this.
>
> (ii) Contradictory Contribution Claims: In the rebuttal, you state: "The reviewer suggests that integrating the physical forward model is our main contribution. We clarify that this is not the case." Yet, contribution list (Line 81) explicitly highlight this integration as "a main contribution". Please be consistent here.
>
> (iii) Related Work and Cohort-Based Learning: Regarding the dismissal of cohort-based learning papers: I find the argument that these are "fundamentally different" because they are super-resolution tasks to be unconvincing in the context of a related work section. I fail to see how meta-learned, *medical image* INRs are not relevant to this paper in the contextualization of this paper.
> Moreover, they also represent commonly used meta-learning frameworks for INRs. Why not contextualize the work of Tancik et al. with respect to other common INR meta learning frameworks in related work?

---

> ### Comment · Reviewer_j28c · 2025-11-27
> **Response to Rebuttal - Part 2**
>
> ### Technical Novelty Concerns
>
> I respectfully disagree with the authors' assessment that my unprecise wording invalidates my broader/general concerns regarding *technical* novelty, and especially *contextualization/presentation*, within the provided references and other work. While the specific application (reconstruction vs. super-resolution) differs, my central claims regarding technical novelty and presentation remain in-tact:
>
> "Unexplored" Claims: The statement that meta-learning INRs in "medical image reconstruction inverse problems" (l.055) "remains unexplored" (l.056) relies on a very narrow definition of the problem setup. While the exact experimental setup (reference-free reconstruction for MRI reconstruction) may be different, the concept of using meta-learning for weight initialization in medical neural fields is not new. A strong (technical) paper should acknowledge this proximity rather than claiming the field is "unexplored".
>
> *Incremental technical novelty*: The proposed method applies/combines an established meta-learning framework for INRs (Tancik et al.) with an established physical model (e.g. Feng et al) to a new domain (MRI reconstruction). I do *NOT* claim incremental *application novelty*, in-fact I complimented the authors on the great work.
>
> ### Summary:
>
> I specifically would like to highlight that I truly believe the authors' work to be of great value to the MRI community, and that its improvements to be substantial. However, since it is an application of an existing meta-learning framework established in MRI, I don't view this as a technical contribution and rather see this to be of greater value for MRI domain experts.  If I consider the publication venues of the references cited, such as [1,2,3], i believe that the paper would be more appreciated at venues specific to MRI reconstruction, as previously pointed out. The authors' strong distinction between MR reconstruction and Super-Resolution, to the extent that SR methods are considered fundamentally different and irrelevant as prior work, reinforces the assessment that this method is tailored to a specific niche application rather than offering a generalized learning representation contribution typically expected at ICLR.
>
>
> [1] Jie Feng, Ruimin Feng, Qing Wu, Zhiyong Zhang, Yuyao Zhang, and Hongjiang Wei. Spatiotemporal implicit neural representation for unsupervised dynamic mri reconstruction. arxiv 2022/TMI'25
>
> [2] Wenqi Huang, Hongwei Bran Li, Jiazhen Pan, Gastao Cruz, Daniel Rueckert, and Kerstin Hammernik. Neural implicit k-space for binning-free non-cartesian cardiac mr imaging. IPMI 2023
>
> [3] Veronika Spieker, Wenqi Huang, Hannah Eichhorn, Jonathan Stelter, Kilian Weiss, Veronika A Zimmer, Rickmer F Braren, Dimitrios C Karampinos, Kerstin Hammernik, and Julia A Schnabel. Iconik: Generating respiratory-resolved abdominal mr reconstructions using neural implicit representations in k-space. MICCAI 2023

---

> ### Comment · Reviewer_j28c · 2025-11-27
> **Response to Rebuttal - Part 3**
>
> Please consider my feedback to revise and improve the presentation of your paper, with respect to positioning it, and delineating its details in the context of medical MRI reconstruction INRs. I am happy to participate in further discussion, and provide feedback on changes.

---

> ### Author Response · Authors · 2025-11-27
>
> We thank the reviewer for the feedback. Below are our detailed responses to the remaining questions.
>
> **Q2: Hyperparameter Sweep: Did you conduct a learning rate hyperparameter sweep for the non-meta-learned INRs? Since meta-learning heavily changes the training dynamics, it would be important to report the best learning rate configuration for the vanilla models, especially since the meta-learning framework has its own optimization loop.**
>
> **A:** We ensure that all learning rates are the best settings for the vanilla models. Table S4 presents our comprehensive hyperparameter sweep for models without IPOD initialization across multiple orders of magnitude. The results demonstrate that our parameter settings achieve optimal performance for each architecture.
> We will include these detailed results in the supplementary materials.
>
> | | | | | | |
> |-|-|-|-|-|-|
> |DINER|LR|5.0e-02|**1.0e-02**|5.0e-03|2.0e-03|
> |DINER|PSNR|46.62±0.16|**46.66±0.14**|46.59±0.14|42.76±3.66|
> |SIREN|LR|2.0e-04|**1.0e-04**|8.0e-05|5.0e-05|
> |SIREN|PSNR|40.23±1.37|**41.42±1.26**|39.55±0.09|35.62±1.22|
> |HASH (Fixed MLP)|LR|7.5e-02|**5.0e-02**|5.0e-03|1.0e-03|
> |HASH (Fixed MLP)|PSNR|46.35±0.01|**46.63±0.15**|46.49±0.04|45.92±0.03|
> |HASH (Fixed Encoder)|LR|5.0e-03|**1.0e-03**|5.0e-04|1.0e-04|
> |HASH (Fixed Encoder)|PSNR|46.26±0.13|**46.63±0.15**|46.50±0.09|46.03±0.06|
>
> *Table S4: Learning rate hyperparameter sweep for INRs without IPOD. Bold indicates optimal configuration.*
>
> ---
>
> **Q3: "Unsupervised" Paradigm: In several sections of the paper (Abstract, Introduction, Methods), the authors present INRs as an "unsupervised paradigm." Could the authors please elaborate on this? I personally believe they are quite the contrary: they overfit to signals, they embed information in their weights (or latents in the context of modulated INRs), and they have a supervised loss (even in the context of k-space INRs).**
>
> **A:** We respectfully maintain "unsupervised" is appropriate, distinguishing our method from supervised learning approaches that require fully-sampled reference data for model training.
>
> **Key distinction:**
> - **Supervised methods:** Require large-scale paired datasets for training models to learn degradation-to-clean mappings
> - **Our unsupervised approach:** Operates solely on acquired undersampled data during reconstruction, without external reference datasets or supervised training
>
> **Established usage:** This definition is standard in INR-based medical imaging [1-6], where "unsupervised" indicates reconstruction without fully-sampled training data:
>
> **Practical significance:** "Unsupervised" emphasizes our method works directly with undersampled measurements without requiring:
> - Large-scale fully-sampled training datasets
> - Paired data collection infrastructure
>
> **Clarification**: While optimization involves data consistency loss (technically "self-supervised" in ML venues), the medical imaging community uses "unsupervised" to distinguish methods by data requirements. We will add a clarifying note in the revision.
>
>
>
> > [1] Catalán, Tabita, et al. "Unsupervised reconstruction of accelerated cardiac cine MRI using neural fields." Computers in Biology and Medicine 185 (2025): 109467.
> >
>
> >[2] Feng, Jie, et al. "Spatiotemporal implicit neural representation for unsupervised dynamic MRI reconstruction." IEEE Transactions on Medical Imaging (2025).
> >
>
> >[3] Wu, Qing, et al. "Moner: Motion Correction in Undersampled Radial MRI with Unsupervised Neural Representation." arXiv preprint arXiv:2409.16921 (2024).
> >
>
> >[4] Tian, Xuanyu, et al. "Unsupervised Motion-Compensated Decomposition for Cardiac MRI Reconstruction via Neural Representation." arXiv preprint arXiv:2511.11436 (2025).
> >
>
> >[5] Liu, Yuanyuan, et al. "Patch-based Reconstruction for Unsupervised Dynamic MRI using Learnable Tensor Function with Implicit Neural Representation." arXiv preprint arXiv:2505.21894 (2025).
> >
>
> >[6] Fan, Kailong, et al. "IMREPET: Implicit Neural Representation for Unsupervised Dynamic PET Reconstruction." International Conference on Medical Image Computing and Computer-Assisted Intervention. Cham: Springer Nature Switzerland, 2025.

---

> ### Author Response · Authors · 2025-11-27
>
> **Q4:Transferability Claim: In lines 098-100, the authors state: "This limitation is a primary reason for their unstable performance and relatively slow reconstruction speeds, as each new dataset requires training from scratch, and the learned representations are difficult to transfer even to similar data domains." Is there any evidence to support the claim that meta-learning makes this process more stable? Why would meta-learning avoid this, and do your experiments validate this claim?**
>
> **A:** Meta-learning improving transferability is well-documented in recent INR works [1,2,3] for computer vision tasks, such as view synthesis, volume encoding, and video generation. These works consistently demonstrate that meta-learned networks exhibit effective generalization to unseen tasks and strong weight transferability across different data distributions, validating that transferability is a fundamental advantage of meta-learning for INR-based methods.
>
> **Our Experimental Validation**
> We conducted comprehensive experiments to validate both transferability and stability claims:
>
> ### **Experiment 1: Cross-dataset Transfer**
> - Setup: Trained only on fastMRI dataset, tested on MoDL dataset.
> - Results are shown in `Figure 3,8,11 and Table 1`
>
> ### **Experiment 2: Cross-contrast transfer**
>   - Setup: Trained only on T2w and FLAIR contrast, tested on T1w contrast.
>   - Results are shown in `Figure 5,9`
>
> ### **Experiment 3: Cross-mask-pattern transfer**
> - Setup: Trained only on Cartesian and Rondom mask patterns, tested on Poisson and Gaussian      mask patterns
> - Results are shown in `Figure 10`
>
> ### **Experiment 4: Cross-forward-model transfer**
> - Setup: Trained only on FFT forward model, tested on NUFFT forward model.
> - Results are shown in `Figure 3,11 and Table 1`
>
> Across all experiments, IPOD initialization offers a powerful prior that accelerates convergence during optimization and enhances generalization capabilities for unseen inverse problems.
> > [1] Chen, Yinbo, and Xiaolong Wang. "Transformers as meta-learners for implicit neural representations." European Conference on Computer Vision. Cham: Springer Nature Switzerland, 2022.
> >
>
> >[2] Yang, Maizhe, Kaiyuan Tang, and Chaoli Wang. "Meta-INR: Efficient Encoding of Volumetric Data via Meta-Learning Implicit Neural Representation." 2025 IEEE 18th Pacific Visualization Conference (PacificVis). IEEE, 2025.
> >
>
> >[3] Guo, Jialong, et al. "MetaNeRV: Meta Neural Representations for Videos with Spatial-Temporal Guidance." Proceedings of the AAAI Conference on Artificial Intelligence. Vol. 39. No. 3. 2025.

---

> > ### Author Response · Authors · 2025-12-01
> >
> > ## Restatement of IPOD's Novelty :
> > IPOD introduces a new inverse-problem-driven meta-learning paradigm for INR that learns a large-scale population prior directly from raw undersampled measurements, without requiring any high-quality images.
> >
> > **Our IPOD novelty lies in two-fold:**
> > - Existing meta-learning methods for INRs generally learn good initialization by constructing image-regression or image-fitting tasks from high-quality images. however, in many computational imaging scenes, **obtaining high-quality ground truth is often impractical or infeasible** due to limitation of acquisition hardware or reconstruction algorithms.
> >
> > - although Existing INR methods for solving inverse problems in computational imaging is promising. They are fundamentally limited by their case-specific optimization manner. Therefore, the model **lacks the ability to effectively leverage the rich prior information shared across different subjects, anatomies, contrasts, or sampling patterns.** This severely restricts the performance and efficiency of current INR approaches.
> >
> > **IPOD addresses both limitations through:**
> >
> > - Reference-free meta-learning: **We learn robust priors directly from raw undersampled measurements** eliminating the dependency on expensive fully-sampled references. Furthermore, the newly proposed adaptive weights mechanism effectively stabilizes the optimization direction of meta-learning.
> >
> > - Population-level prior learning: Through meta-learning across diverse inverse problems set (anatomies, sampling patterns, acceleration factors), **IPOD captures generalizable initialization that transfers effectively to unseen subjects and protocols**, fundamentally distinct from meta-learning framework established in computational imaging literature.

---

> > > ### Author Response · Authors · 2025-12-01
> > >
> > > ## On Expansion of related work:
> > >
> > > - In our original submission, the k-space recon INRs and image-space recon INRs were not explicitly distinguished due to space limitations. In the revised submission, we have provided the clear categorization of these two paradigms in the subsection "INR for MRI Reconstruction" of `Sec. 2`.
> > >
> > > - In the revised submission, we have incorporated the discussion of existing meta-learning frameworks for medical imaging in the subsection "Meta-Learning" of `Sec. 2`. This expanded discussion clarifies how IPOD differs from these approaches while acknowledging their relevance as prior work in meta-learning for medical imaging INRs.

---

### Official Review · Reviewer_26Vt · 2025-10-31

**Soundness:** 2
**Presentation:** 3
**Contribution:** 2
**Rating:** 4
**Confidence:** 3

**Summary:**

This paper introduces IPoD a meta learning method for initializing INR weights from data to be used downstream in reconstruction tasks. The authors show that by utilizing their meta-learning initialization for INRs they can achieve higher quality image reconstructions with INRs than with random initialized INRs. Importantly they show that this method of initialization only requires access to under sampled data which is an important when training inverse problems in the self-supervised setting.  The main results of the paper show that both visually and numerically that their initialization outperforms random initialization.  They showed that their method worked on a variety of different datasets, and most importantly, on prospectively collected MR data.

**Strengths:**

The paper provides an interesting method for improving the performance of INR based image reconstruction methods. The experiments show nice performance gains in the reconstruction quality over random initialization which are not just numerical but are also clearly visible in the reconstructions. Additionally, its really great to see results on prospectively collected data which is seldom shown for new reconstruction techniques.

**Weaknesses:**

I do believe that there are comparisons to existing work which should be included. As this method is a self-supervised approach the authors should compare to at least one other non-INR self-supervised recon technique like "Self-Supervised Learning of Physics-Guided
Reconstruction Neural Networks without FullySampled Reference Data". Although this method is not INR based, there are existing popular self-supervised methods for MRI reconstruction that are even used on some of the same datasets. On top of this, I would like to see metrics for different acceleration levels on the 1D cartesian under sampling example. If the authors are dealing with multi-coil data, R=3,4 is not too challenging of a task. Typically even parallel imaging + hand crafted regularization can do very well here. Please try and include some metrics for at least R=8 1D cartesian sampling if possible or compare to hand crafted regularization techniques at R=3,4. Without these additional comparisons it is difficult to conclude if the method provides improvements to the broader category of self-supervised reconstruction techniques.

**Questions:**

1. How does the method preform at higher acceleration levels?
2. How does the method compare to existing self-supervised reconstruction techniques like SSDU?
3. How long does each reconstruction take in seconds?

---

> ### Author Response · Authors · 2025-11-22
>
> We sincerely appreciate your time and effort in reviewing our work. Your constructive feedback motivates us to further improve our study. Below, we address your comments in detail.
>
> **Q1: How does the method perform at higher acceleration levels?**
>
> **Q2: How does the method compare to existing self-supervised reconstruction techniques like SSDU?**
>
> **A (Joint Response to Q1 and Q2):** Thank you for your important questions and constructive suggestions. We address both together by conducting comprehensive comparisons at multiple acceleration factors, which are shown in our supplementary.
>
> **Experimental Setup:** To thoroughly evaluate IPOD's performance, we compared against:
>
> - **Hand-crafted regularization techniques:** Total Variation (TV) and L1 Wavelet
>
> - **Self-supervised method:** SSDU [1]
>
> We evaluated on 1D Cartesian undersampling at AF=4 and AF=8, representing moderate to challenging acceleration factors widely used in accelerated MRI.
>
> **Implementation details:**
>
> - **Coil sensitivity maps:** All experiments used identical coil sensitivity maps pre-computed from the ESPIRiT method
>
> - **SSDU baseline:** We adhered to the protocol in [1] with identical network architectures and hyperparameters. The training dataset matches that used in IPOD meta-learning
>
> - **Fair comparison:** The pre-trained SSDU model was carefully fine-tuned for 500 iterations on test data to ensure optimal performance
>
> **Results:**
>
> The visualized and quantitative results are shown in `Figure S3 and Table S3 in our supplementary`. The analysis of results are as follows:
>
> **- Quantitative performance:** As shown in Table S3, All IPOD-based methods achieve superior reconstruction quality compared to both hand-crafted and self-supervised baselines in both PSNR and SSIM metrics across both acceleration factors.
>
> **- Qualitative analysis:** Figure S3 demonstrates that IPOD-based methods effectively suppress aliasing artifacts in challenging anatomical regions (yellow arrows in zoom-in panels), which remain prominent in TV, L1 Wavelet, and SSDU reconstructions.
>
> These results confirm that IPOD remains effective in 1D Cartesian undersampling with different acceleration factors, compared with hand-crafted regularization techniques and SSDU.
>
> | AF | Metric | TV | L1Wavelet | SSDU | DINER-IPOD | SIREN-IPOD | HASH-IPOD |
> |:--:|:------:|:------:|:---------:|:------:|:----------:|:----------:|:---------:|
> | 4 | PSNR | 43.54 ± 3.12 | 44.66 ± 0.73 | 45.33 ± 0.56 | **46.81 ± 1.14** | 46.55 ± 0.02 | 46.67 ± 0.02 |
> | 4 | SSIM | 0.983 ± 0.016 | 0.988 ± 0.008 | 0.993 ± 0.005 | 0.997 ± 0.002 | 0.998 ± 0.009 | 0.998 ± 0.007 |
> | 8 | PSNR | 27.36 ± 3.90 | 29.19 ± 1.11 | 35.72 ± 1.61 | 36.66 ± 2.51 | **36.81 ± 1.37** | 36.68 ± 2.62 |
> | 8 | SSIM | 0.927 ± 0.013 | 0.928 ± 0.006 | 0.938 ± 0.011 | 0.943 ± 0.005 | 0.940 ± 0.001 | 0.947 ± 0.005 |
>
> *Table S3: Quantitative comparison (Mean ± Std in PSNR/SSIM) of hand-crafted regularization techniques, SSDU and IPOD (with different network backbones) for 1D Cartesian undersampling.*
>
> > [1] Yaman B, Hosseini SAH, Moeller S, Ellermann J, Uğurbil K, Akçakaya M. Self-supervised learning of physics-guided reconstruction neural networks without fully sampled reference data. Magn Reson Med. 2020 Dec;84(6):3172-3191.
>
> ---
>
> **Q3: How long does each reconstruction take in seconds?**
>
> **A:** We compared the reconstruction time consumption among the hand-crafted regularization techniques, SSDU and the INR-based methods with and without IPOD (`Figure S4 and Table S4 in our supplementary`).
>
> **Experimental Setup:**
> - **Hardware:** NVIDIA A100 GPU
> - **Image size:** 256×256 per 2D slice.
> - **Convergence criterion:** PSNR stabilization with change rate < 0.01 dB/iteration
> - **Reported time:** Pure reconstruction time (excluding data I/O), averaged across Cartesian (AF = 4), Gaussian (AF = 6) and Poisson (AF = 8) undersampling patterns.
>
> Table S4 shows that IPOD significantly accelerates reconstruction speed across all INR backbones by 2×~3×. Notably, HASH-IPOD (4.2s) matches the approximate speed of handcrafted regularization methods (L1Wavelet: 3.5s), and all IPOD-initialized methods reconstruct faster than SSDU.
>
> | Method | w/o IPOD | w/ IPOD |
> |--------|----------|---------|
> | L1 Wavelet | 3.5 | - |
> | TV | 15.3 | - |
> | SSDU | 42.6 | - |
> | DINER | 13.5 | **5.8** |
> | SIREN | 116.7 | **37.8** |
> | HASH | 11.6 | **4.2** |
>
> *Table S4: The average reconstruction time comparison of hand-crafted regularization methods, SSDU, and IPOD (with different network backbones) across Cartesian, Gaussian, and Poisson undersampling patterns.*

---

### Official Review · Reviewer_Mi8r · 2025-11-01

**Soundness:** 3
**Presentation:** 4
**Contribution:** 3
**Rating:** 8
**Confidence:** 4

**Summary:**

This paper introduces IPOD (Inverse-Problem-Driven Meta-Learning), a novel meta-learning framework designed to find generalizable parameter initializations for Implicit Neural Representations (INRs) in accelerated MRI reconstruction. IPOD leverages physics-informed optimization across diverse, undersampled reconstruction tasks, eliminating the need for fully-sampled ground truth images during meta-training and achieving faster convergence and superior reconstruction quality.

**Strengths:**

1. The framework consistently achieves faster convergence and superior reconstruction quality across diverse, unseen out-of-domain scenarios, including different anatomies, contrast mechanisms, and sampling patterns/protocols. This is a major improvement over conventional scan-specific INR methods that suffer from slow speeds.

2. IPOD is shown to be a unified framework that provides effective initialization for multiple distinct INR architectures (DINER, SIREN, HASH).

3. The performance is consistently better than all the baseline methods.

4. The writing and figures are clear and easy to follow.

**Weaknesses:**

1. The study primarily focuses on 2D MRI reconstruction tasks. The application to 3D MRI reconstruction is unexplored.

2. A  missing ablation study is proof for the paper's claim that its diverse problem set (e.g., varying sampling patterns and anatomies) creates a robust prior. The authors should have compared their model's generalization against models meta-trained on non-diverse, specialized sets (e.g., only brains or only Cartesian sampling) to quantify the actual benefit of this diversity.

**Questions:**

Same as the weakness part.

---

> ### Author Response · Authors · 2025-11-22
>
> Thank you for the insightful comments. We are encouraged by your recognition of our work. Below, we provide detailed point-by-point responses to address your concerns.
>
> **W1:** The study primarily focuses on 2D MRI reconstruction tasks. The application to 3D MRI reconstruction is unexplored.
>
> **A:** We appreciate this observation and clarify that while our experiments demonstrate slice-wise reconstruction, **IPOD can be directly applied to 3D MRI reconstruction** without modification.
>
> We are currently conducting 3D experiments with slice-wise processing across multiple undersampling patterns, with results forthcoming in the supplementary.
>
> **Why 2D Framework Applies to 3D MRI:** In clinical 3D MRI acquisition, it is common to fully sample one spatial dimension—typically the Cartesian readout direction, often aligned with the z-axis[1-4]. When the $k_z$ dimension is fully sampled, a 1D FFT along $k_z$ converts the 3D k-space data into a series of independent 2D k-space data in the $k_x$-$k_y$ plane. This slice-wise decomposition is standard in 3D MRI reconstruction workflows and applies even to non-Cartesian trajectories such as stack-of-stars and stack‐of‐spirals [5-7]. The final 3D volume is then obtained by reconstructing each 2D slice separately and stacking the results along the z-axis.
>
> However, for certain 3D non-Cartesian sampling patterns (e.g., 3D cones or Kooshball trajectories [8]), performing a slice-wise decomposition along any axis is non-trivial. Extending IPOD to these inherently 3D sampling schemes would therefore require additional designs, and we leave this direction for future work. Nonetheless, such acquisition patterns remain relatively niche compared with routine clinical protocols (e.g., Cartesian or stack-based trajectories), and our current 2D-based formulation already covers the majority of practical 3D MRI use cases.
>
> > [1] Dong, Zijing, et al. "Variable flip angle echo planar time-resolved imaging (vFA-EPTI) for fast high-resolution gradient echo myelin water imaging." Neuroimage 232 (2021): 117897.
> >
>
> > [2] Wang, Fuyixue, et al. "3D Echo Planar Time-resolved Imaging (3D-EPTI) for ultrafast multi-parametric quantitative MRI." Neuroimage 250 (2022): 118963.
> >
>
> > [3] Jung, Woojin, et al. "Highly accelerated 3D MPRAGE using deep neural network–based reconstruction for brain imaging in children and young adults." European Radiology 32.8 (2022): 5468-5479.
> >
>
> > [4] Lao, Guoyan, et al. "Coordinate-based neural representation enabling zero-shot learning for fast 3D multiparametric quantitative MRI." Medical Image Analysis 102 (2025): 103530.
> >
>
> > [5] Zhou, Ziwu, et al. "Golden‐ratio rotated stack‐of‐stars acquisition for improved volumetric MRI." Magnetic resonance in medicine 78.6 (2017): 2290-2298.
> >
>
> > [6] Chang, Yulin V., et al. "3D‐accelerated, stack‐of‐spirals acquisitions and reconstruction of arterial spin labeling MRI." Magnetic resonance in medicine 78.4 (2017): 1405-1419.
> >
>
> > [7] Zhang, Xiaoyong, et al. "3D self‐gated cardiac cine imaging at 3 tesla using stack‐of‐stars bSSFP with tiny golden angles and compressed sensing." Magnetic Resonance in Medicine 81.5 (2019): 3234-3244.
> >
>
> > [8] Ding, Zekang, et al. "Reduction of ringing artifacts induced by diaphragm drifting in free‐breathing dynamic pulmonary MRI using 3D koosh‐ball acquisition." Magnetic Resonance in Medicine 92.5 (2024): 2021-2036.

---

> > ### Author Response · Authors · 2025-12-01
> >
> > Following the reviewer's valuable suggestion, we have now completed comprehensive 3D MRI reconstruction experiments. We acquired three fully-sampled, high-resolution (1×1×1 mm³) 3D brain datasets and performed retrospective undersampling reconstruction studies with various acceleration factors and sampling patterns. The detailed qualitative and quantitative comparisons are shown in `Figure S6 and Table S5 (supplementary)`.
> >
> > The results demonstrate that benefiting from the learned robust prior, IPOD-reconstructed images preserve superior structural details and effectively suppress noise interference, which is particularly severe in high-resolution acquisitions.
> >
> > | Sampling | AF | Metric | w/ IPOD | w/o IPOD |
> > |----------|:--:|:------:|:-------:|:--------:|
> > | 1D Cartesian | 4 | PSNR | **35.21 ± 2.09** | 34.33 ± 1.96 |
> > | 1D Cartesian | 4 | SSIM | **0.985 ± 0.008** | 0.971 ± 0.009 |
> > | 2D Poisson | 6 | PSNR | **33.21 ± 1.37** | 31.52 ± 2.09 |
> > | 2D Poisson | 6 | SSIM | **0.978 ± 0.007** | 0.964 ± 0.009 |
> > | 2D Gaussian | 8 | PSNR | **31.60 ± 1.49** | 29.78 ± 1.33 |
> > | 2D Gaussian | 8 | SSIM | **0.9626 ± 0.014** | 0.9555 ± 0.013 |
> >
> > *Table S5: Quantitative comparison (Mean ± Std in PSNR/SSIM) of 3D reconstruction results with and without IPOD across different acceleration factors and sampling patterns.*

---

> ### Author Response · Authors · 2025-11-22
>
> **W2:** A missing ablation study is proof for the paper's claim that its diverse problem set (e.g., varying sampling patterns and anatomies) creates a robust prior. The authors should have compared their model's generalization against models meta-trained on non-diverse, specialized sets (e.g., only brains or only Cartesian sampling) to quantify the actual benefit of this diversity.
>
> **A:** To evaluate the effect of problem set diversity, we have conducted the exact ablation study requested to demonstrate that the diverse problem set provides robust priors.
>
> **Experimental Setup:** We compared three meta-learning strategies:
>
> - **IPOD-Brain:** The problem set in IPOD learning phase only specialized on brain anatomy data.
>
> - **IPOD-Cartesian:** The problem set in IPOD learning phase only specialized on data from Cartesian undersampling.
>
> - **IPOD-All:** The whole problem set (multiple anatomies + undersampling patterns) used in IPOD learning phase.
>
> Note that all the hyperparameters in these three strategies remained consistent with those reported in the main paper. We compare the prior generalization of IPOD-Cartesian and IPOD-All on radial undersampled data (`Figure S1 and Table S1 in supplementary`), while comparing IPOD-Brain and IPOD-All on knee data (`Figure S2 and Table S2 in supplementary`).
>
> **Key Observations:**
>
> - IPOD-All achieves the best performance in all tests with faster convergence.
>
> - IPOD-Brain and IPOD-Cartesian at the early iteration (100 iterations) present artifacts and blurring of structural details, which are not observed in IPOD-All.
>
> - All these three strategies offer a prior that accelerates the network adapting on the unseen data compared with that without initialization from IPOD.
>
> The fact that IPOD-All outperforms IPOD-Brain on knee data and outperforms IPOD-Cartesian on radial undersampled data directly proves our central claim: exposure to diverse inverse problems during IPOD meta-learning prevents prior overfitting to specific problem characteristics and produces more robust, generalizable initialization.
>
> We will add this ablation study to the revised manuscript as requested.
>
> | Method | Metric | Iter 10 | Iter 100 | Iter 500 | Iter 1000 |
> |--------|--------|---------|----------|----------|-----------|
> | w/ IPOD | PSNR | 12.74 ± 0.98 | 23.90 ± 1.87 | 29.17 ± 0.31 | 31.47 ± 0.45 |
> | | SSIM | 0.510 ± 0.056 | 0.772 ± 0.006 | 0.891 ± 0.017 | 0.934 ± 0.006 |
> | IPOD-Cartesian | PSNR | 16.40 ± 4.36 | 26.80 ± 0.84 | 31.91 ± 0.25 | 32.92 ± 0.79 |
> | | SSIM | 0.530 ± 0.017 | 0.832 ± 0.011 | 0.927 ± 0.002 | 0.953 ± 0.008 |
> | **IPOD-All** | **PSNR** | **18.51 ± 1.00** | **30.97 ± 0.99** | **33.78 ± 1.76** | **34.21 ± 2.96** |
> | | **SSIM** | **0.604 ± 0.022** | **0.907 ± 0.009** | **0.957 ± 0.007** | **0.968 ± 0.006** |
>
> *Table S1: Quantitative comparison (Mean ± Std in PSNR/SSIM) of IPOD with different meta-learning strategies across progressive iterations for radial undersampling. (IPOD-Cartesian: meta-trained on Cartesian undersampling only; IPOD-All: meta-trained on diverse undersampling patterns.)*
>
> | Method | Metric | Iter 10 | Iter 100 | Iter 500 | Iter 1000 |
> |--------|--------|---------|----------|----------|-----------|
> | w/o IPOD | PSNR | 13.81 ± 2.62 | 26.59 ± 3.11 | 29.99 ± 2.81 | 31.71 ± 2.83 |
> | | SSIM | 0.539 ± 0.057 | 0.833 ± 0.050 | 0.904 ± 0.026 | 0.929 ± 0.018 |
> | IPOD-Brain | PSNR | 17.74 ± 1.08 | 29.33 ± 3.53 | 31.91 ± 3.68 | 33.14 ± 3.38 |
> | | SSIM | 0.667 ± 0.042 | 0.899 ± 0.040 | 0.937 ± 0.028 | 0.950 ± 0.023 |
> | **IPOD-All** | **PSNR** | **19.57 ± 2.58** | **30.72 ± 3.35** | **33.04 ± 3.93** | **34.74 ± 3.58** |
> | | **SSIM** | **0.726 ± 0.044** | **0.919 ± 0.029** | **0.951 ± 0.020** | **0.960 ± 0.017** |
>
> *Table S2: Quantitative comparison (Mean ± Std in PSNR/SSIM) of IPOD with different meta-learning strategies across progressive iterations on fastMRI Knee Dataset. (IPOD-Brain: meta-trained on brain dataset only; IPOD-All: meta-trained on diverse anatomies.)*

---

### Author Response · Authors · 2025-12-03
**Summary of Revisions and Gentle Reminder**

Dear AC and Reviewers,

**We deeply appreciate the insightful feedback, which has significantly improved our work. Below is a summary of the revisions we have made in response to your suggestions**

1. Evaluating IPOD effectiveness on 3D MRI reconstruction with various sampling patterns and acceleration factors (`Figure S6, Table S5`)[Mi8r].

2. Demonstrating that problem set diversity is crucial for learning robust priors in reference-free meta-learning (`Figures S1-S2, Tables S1-S2`) [Mi8r].

3. Showing IPOD outperforms hand-crafted regularization and self-supervised methods at moderate to challenging acceleration factors (`Figure S3, Table S3`) [26Vt].

4. Providing detailed reconstruction time comparison, where IPOD achieves 2-3× speedup across all INR architectures (`Figure S4, Table S4`) [26Vt].

5. Providing more comprehensive convergence curve analysis, where IPOD achieves faster convergence and superior reconstruction quality (`Figure S5`) [j28c].

6. Demonstrating fair comparison with vanilla INR networks via comprehensive hyperparameter sweep (`Table S6`) [j28c].

**We believe these revisions substantially address all the concerns raised and further demonstrate the superiority and robustness of the IPOD framework.**

Best regards,

The authors

---

> ### Author Response · Authors · 2025-12-03
> **Restatement of IPOD's Novelty**
>
> In light of reviewer j28c's acknowledged misunderstanding regarding technical aspects (as clarified in discussion) and comments regarding novelty, we would like to further restate and highlight the key innovations of IPOD.
>
> ## IPOD proposes a reference-free meta-learning framework for effective inverse problem solving:
>
> **Reference dependency:** Existing meta-learning methods for INRs learn good initialization by constructing image-regression from expensive fully-sampled references, which is often impractical or infeasible in many computational imaging scenarios.
>
> **Our IPOD meta-learns directly from undersampled raw measurements** by leveraging the physical forward model and a novel adaptive weighting mechanism. Comprehensive results are provided in `Figures 3-7, Tables 1-2, and Tables S1-S5`.
>
> ## IPOD enhances INR-based framework performance by utilizing population-level priors:
>
> **Neglecting shared priors:** Conventional INRs for inverse problems are fundamentally limited by their case-specific optimization manner, and lack the ability to effectively leverage the rich prior information shared across different scenarios.
>
> **Our IPOD meta-learns population-level priors from diverse problem sets**, capturing generalizable initialization that transfers effectively to unseen subjects and protocols. Comprehensive validation is provided in `Figures 3-5, 8-9, and Tables S1-S3`.

---

### Note · Authors · 2026-01-28

I have read and agree with the venue's withdrawal policy on behalf of myself and my co-authors.

---

### Meta-Review · Area_Chair_Fugn · 2026-01-07

**Summary:**

This work introduces a meta-learning framework that learns generalizable model initializations for implicit neural representations (INRs) for undersampled MRI reconstruction. By integrating the physics-based forward model directly into the meta-learning process, this model enables INRs to achieve faster convergence and improved reconstruction quality compared to baselines. A key advantage is that the model requires only undersampled data for meta-training, eliminating the need for fully sampled ground-truth images in practical applications.

Reviewers raised the concerns about technical contributions, problem and experiment setting, comparison methods, presentation structures especially in related work about positioning the work. The author provides a detailed response in rebuttal with newly added experiment results including comparison with SSDU, ablation studies with different priors, and hyperparameter tuning analysis. One reviewer responded and followed up with a detailed discussion and suggestions to improve the paper, which is very appreciated. The rebuttal helps to clarify and answer quite a few questions while there may still be some remaining concerns. Specifically, I share the comment from Reviewer j28c on the technical novelty. I appreciate the key point for meta-training with undersampled data only while not fully convinced with all the claims by the author in rebuttal. With prior work already exploring INRs for MRI reconstruction and meta-training for improved initialization, this paper clearly takes a meaningful step forward, while it may be helpful to further elaborate on the technical challenges addressed in this extension. The motivation of removing the need for ground-truth data in meta learning is great, though additional thinking on whether 2D MRI reconstruction is the most compelling application might be helpful, given publicly available datasets and small scale of required training data. Highlighting scenarios such as high-dimensional dynamic MRI or other scientific imaging problems where full-sampled data or ground truth is naturally unavailable might make the contribution and unique advantage even more convincing. Overall, the authors may find some of the reviewers’ comments useful for further refining and enhancing the manuscript.

**Reviewer Concerns:**

Reviewers raised the concerns about technical contributions, problem and experiment setting, comparison methods, presentation structures especially in related work about positioning the work. The author provides a detailed response in rebuttal with newly added experiment results including comparison with SSDU, ablation studies with different priors, and hyperparameter tuning analysis. One reviewer responded and followed up with a detailed discussion and suggestions to improve the paper, which is very appreciated. The rebuttal helps to clarify and answer quite a few questions while there may still be some remaining concerns. Specifically, I share the comment from Reviewer j28c on the technical novelty. I appreciate the key point for meta-training with undersampled data only while not fully convinced with all the claims by the author in rebuttal. With prior work already exploring INRs for MRI reconstruction and meta-training for improved initialization, this paper clearly takes a meaningful step forward, while it may be helpful to further elaborate on the technical challenges addressed in this extension. The motivation of removing the need for ground-truth data in meta learning is great, though additional thinking on whether 2D MRI reconstruction is the most compelling application might be helpful, given publicly available datasets and small scale of required training data. Highlighting scenarios such as high-dimensional dynamic MRI or other scientific imaging problems where full-sampled data or ground truth is naturally unavailable might make the contribution and unique advantage even more convincing. Overall, the authors may find some of the reviewers’ comments useful for further refining and enhancing the manuscript.

**Reviewer Scores:**

Only one out of three reviewers responded to the response. While it is hard to predict the feedback from other two reviewers, this reviewer give a very detailed and comprehensive response in the follow-up discussion to share the remaining concern.

---

### Decision · Program_Chairs · 2026-01-26

Reject